

# Strength and limits of transient mid to late Holocene simulations with dynamical vegetation

Pascale Braconnot*, Dan Zhu, Olivier Marti and Jérôme Servonnat

IPSL/Laboratoire des Sciences du Climat et de l'Environnement, unité mixte CEA-CNRS-UVSQ, Université Paris Saclay, Bât. 714, Orme de Merisiers, 91191 Gif-sur-Yvette Cedex.

*Correspondance to* : pascale.braconnot@lsce.ipsl.fr

**Abstract.** We discuss here the first 6000 years long Holocene simulations with fully interactive vegetation and carbon cycle with the IPSL Earth system model. It reproduces the long term trends in tree line in northern hemisphere and the southward shift of Afro-Asian monsoon precipitation in the tropics in response to orbital forcing. The simulation is discussed at the light of a set of mid Holocene and pre industrial simulations performed to set up the model version and to initialize the dynamical vegetation. These sensitivity experiments remind us that model quality or realism is not only a function of model parameterizations and tuning, but also of experimental set up. They also question the possibility for bi-stable vegetation states under modern conditions. Despite these limitations the results show different timing of vegetation changes through space and time, mainly due to the pace of the insolation forcing and to internal variability. Forest in Eurasia exhibits changes in forest composition with time as well as large centennial variability. The rapid increase of atmospheric $CO_2$ in the last centuries of the simulation contributes to enhance tree growth and counteracts the long term trends induced by Holocene insolation in the northern hemisphere. A complete evaluation of the results would require being able to properly account for systematic model biases and, more important, a careful choice of the reference period depending on the scientific questions.

## 1    Introduction

Past environmental records such as lake levels or pollen records highlight substantial changes in the global vegetation cover during the Holocene (COHMAP-Members, 1988; Wanner et al., 2008). The early to mid-Holocene optimum period was characterized by a northward extension of boreal forest over north Eurasia and America which attests for increased temperature in mid to high latitudes (Prentice and Webb, 1998). The early to mid-Holocene has also seen a massive expansion of moisture and precipitation in Afro-Asian regions that have been related to enhance boreal summer monsoon (Jolly et al., 1998; Lezine et al., 2011). These changes were triggered by latitudinal and seasonal changes in top of the atmosphere (TOA) incoming solar radiation caused by the long term variation in Earth's orbital parameters (Berger, 1978). During the course of the Holocene these features retreated towards their modern distribution (Wanner et al., 2008). While global data syntheses exist for the mid-Holocene (Bartlein et al., 2011; Harrison, 2017; Prentice et al., 2011), reconstructions focus in general on a location or a region when considering the whole Holocene. For example regional syntheses for long term paleo records over Europe reveal long term vegetation changes that can be attributed to changes in temperature or precipitation induced by insolation changes (Davis et al., 2003; Mauri et al., 2015). Similarly, over West Africa or Arabia, pollen data suggests a southward retreat of the intertropical convergence zone (Lezine et al., 2017), and a reduction Africa monsoon intensity (Hély and Lézine, 2014). The pace of these





changes varies from one region to the other (e.g. Fig. 6.9 in Jansen et al., 2007) (Renssen et al., 2012) and has been punctuated by millennium scale variabilityor abrupt events (deMenocal et al., 2000), for which it is still unclear that they represent global or more regional events. How vegetation changes have been triggered by this long term climate change and what has been the vegetation feedback on climate is still a matter of debate.

Pioneer simulations with asynchronous climate-vegetation coupling suggested that vegetation had a strong role in amplifying the African monsoon (Braconnot et al., 1999; Claussen and Gayler, 1997; de Noblet-Ducoudre et al., 2000; Texier et al., 1997). When dynamical vegetation model where included in fully coupled ocean-atmosphere-sea-ice models, climate simulations suggested a lower magnitude of the vegetation feedback (Braconnot et al., 2007a; Braconnot et al., 2007b; Claussen, 2009). Individual model results suggest however that vegetation plays a role in triggering the African monsoon during mid-Holocene (Braconnot and Kageyama, 2015), but also that soil moisture might play a larger role than anticipated (Levis et al., 2004). Dust has also been identified as an important player with dust emission tied to vegetation cover and slow evolution of soil properties (Albani et al., 2015; Egerer et al., 2017; Pausata et al., 2016). In high latitude also the role of the vegetation feedback is not fully understood. Previous studies showed that the response of vegetation in spring combined to the response of the ocean in autumn were key factors to transform the seasonally varying insolation forcing into an annual warming (Wohlfahrt et al., 2004). The magnitude of this feedback has been questioned by Otto et al. (Otto et al., 2009),  showing that vegetation was mainly responding to ocean and sea-ice induced warming over land. The role and magnitude of the vegetation feedback was also questioned over Asia (Dallmeyer et al., 2010). The variety of response of dynamical vegetation models to external forcing is also an issue in these discussions, even though the fact that they all produce increased vegetation in Sahel when forced with mid-Holocene suggest that despite the large uncertainties robust basic response can be inferred from current models (Hopcroft et al., 2017).  Other studies have also highlighted that there might exist several possible vegetation distribution at the regional scale for a given climate that can be related to instable vegetation states (e.g. Claussen, 2009). This is still part of the important questions to solve to fully explain the end of the African humid period around 4000-5000 years BP (Liu et al., 2007).

It is not clear yet that more comprehensive models and long Holocene simulations can help solve all the questions, given all the uncertainties described above. But they can help solve the question of vegetation-climate state and of the linkages between insolation, trace gas forcing, climate and vegetation changes contrasting the evolution between polar, temperate and tropical regions. For this, we investigate the long term trend and variability of vegetation characteristics as simulated by a version of the IPSL model with a fully interactive carbon cycle and dynamical vegetation, considering the last 6000 years.  Previous studies clearly highlight that small differences in the albedo or soil formulation can have large impact on the simulated results (Bonfils et al., 2001; Otto et al., 2011). Given all the interactions in a climate system, the climatology produced by a model version with interactive vegetation is by construction different from the one of the same model with prescribed vegetation. In particular model biases are in general larger (Braconnot and Kageyama, 2015; Braconnot et al., 2007b), so that the corresponding simulations need to be considered as resulting from different models (Kageyama et al., 2018). In this study, we started from the IPSLCM5A-MR version of the IPSL model (Dufresne et al., 2013) and implemented an intermediate version of the land-surface model ORCHIDEE between the one used in IPSLCM5A-MR and the one now included in the IPSLCM6A-LR version of the model. Small tuning and changes in the way we consider the aerosol forcing in the simulation also affects the results of the simulated



mid-Holocene climate, and thereby the transient Holocene simulation with dynamical vegetation. Because of
this, the initial mid-Holocene (6ka BP or MH in the following) starting point with this model cannot be directly
compared to the mid-Holocene simulations ran as part of PMIP3-CMIP5 (Kageyama et al., 2013a). It is
important to know how these changes affect model results and the realisms we can expect from the transient
simulations. We thus investigate first how the different changes we made affect mid-Holocene simulations.
Different strategies can be used to initialize the vegetation dynamics and produce the mid-Holocene initial state
for the transient simulation. We investigate if they have an impact on the simulated vegetation states and if the
transient simulation produces climate and vegetation states compatible with what is obtained from snap shot
experiments. For the transient experiments, the focus will be on the long term trends in climate and vegetation so
as to isolate the direct response to insolation and trace gases forcing. Key questions concern the differences
between hemispheric variations and regional characteristics, considering the timing or the magnitude of the
response to forcings compared to the magnitude of centennial internal variability.

The remainder of the manuscript is organized as follow. The first part describes the model version and

the characteristics of the land surface model we have implemented to account for the dynamical vegetation.
Section 2 discusses possible differences in model initial state depending on the modelling and experimental
choices we made. Section 3 analyses mid Holocene snapshot simulations and the impact of model physics, and
discusses the choice of an initial state for the transient simulation. Section 4 presents the transient simulation
focusing on long term climate and vegetation trends at global and regional scales, before the conclusion in
section 5.

## 2    Model, mid Holocene and preindustrial experiments

### 2.1    The IPSL Earth System Model

We use a modified version of the IPSL model compared to the one used for CMIP5 simulations

(Dufresne et al., 2013). It has the same resolution and the same atmosphere, ocean and sea-ice physics than the
IPSLCM5A-MR model. This model version thus couples the LMDZ.4 atmospheric model with 144x142 grid
points in latitude and longitude (2.5°x1.27°) and 39 vertical levels (Hourdin et al., 2013) to the ORCA2 ocean
model at 2° resolution (Madec, 2008). The ocean grid is such that resolution is enhanced around the equator and
in the Arctic due to the grid stretching and pole shifting. The LIM2 sea-ice model is embedded in the ocean
model to represent sea ice dynamics and thermodynamics (Fichefet and Maqueda, 1999). The ocean
biogeochemical model PISCES is also coupled to the ocean physics and dynamics to represent the marine
biochemistry and the carbon cycle (Aumont and Bopp, 2006). The atmosphere-surface turbulent fluxes are
computed taking into account fractional land-sea area in each atmospheric model grid box. The sea fraction in
each atmospheric grid box is imposed by the projection of the land-sea mask of the ocean model on the
atmospheric grid, allowing for a perfect conservation of energy (Marti et al., 2010). Ocean-sea-ice and
atmosphere are coupled once a day through the OASIS coupler (Valcke, 2006). The land surface scheme is the
ORCHIDEE model (Krinner et al., 2005). It is coupled to the atmosphere at each atmospheric model 30mn
physical time steps and includes a river runoff scheme to route runoff to the river mouths or to coastal areas
(d'Orgeval et al., 2008). Over the ice sheet water is also routed to the ocean and distributed over wide areas so as
to mimic iceberg melting and to close the water budget (Marti et al., 2010). This model accounts for a mosaic



vegetation representation in each grid box, considering 13 (including 2 crops) plant functional types (PFT) and
fully interactive carbon cycle (Krinner et al., 2005).
Compared to the standard version of the IPSLCM5A model described above, several changes were
included in the land-surface model. The first one concerns the inclusion of the 11 layers physically-based
hydrological scheme (de Rosnay et al., 2002) that replaces the 2 layers bucket-type hydrology (Ducoudré et al.,
1993). Several model adjustments had to be done to set up the model version with the 11 layer hydrology
(simulation L11, Table 1). The land surface components were available, but had never been fully tested in the
full coupled mode before this study. We gave specific care to the closure of the water budget of the land surface
model to ensure that $O(1000$ years) simulations will not exhibit spurious drift in sea level. In addition the new
prognostic snow model was included (Wang et al., 2013). The scheme describes snow with 3 layers that are
distributed so that the diurnal cycle and the interaction between snowmelt and runoff are properly represented. In
order to avoid snow accumulation on some grid points, snow depth is not allowed to exceed 3m. The excess
snow is melted and included in soil and runoff while conserving water and energy (Charbit and Dumas, pers.
communication). Because of a large cold bias in high latitudes in the first tests, we also reduced the bare soil
albedo that is used to combine fresh snow and vegetation in the snow aging parameterization.
The version of the model used for the transient late Holocene simulation also accounts for the changes
in vegetation in response to climate and $CO_2$ evolution. Off line simulations, using the original scheme for
dynamical vegetation of ORCHIDEE, were already used to analyze Mid-Holocene and LGM vegetation forced
with climate simulated by the IPSLCM5A-LR model (Kageyama et al., 2013b; Woillez et al., 2011). Here we
switch on the dynamical vegetation model described in Zhu et al. (2015). Compared to the original scheme
(Krinner et al., 2005), this version of the land surface model produces more realistic vegetation distribution in
mid and high latitude regions when compared with present-day observations. We conducted several tests to
initialize the vegetation distribution for this first long mid to late Holocene transient simulation as discuss in
section 3.

### 2.2    Mid Holocene experimental design

The mid-Holocene (MH) time-slice climate experiment (6000 years BP) represents the initial state for
the transient late Holocene simulation with dynamical vegetation. It is thus considered as a reference climate in
this study. Because of this, and to save computing time, all model adjustments made to set up the model content
and the model configuration were mainly done using mid-Holocene simulations and not pre-industrial
simulations. Only a subset of tests is available for the pre-industrial period as shown in Table 1 and 2.
The MH simulations have been performed with Earth's orbit and trace gazes prescribed to the 6kyr BP
conditions. Compared to previous PMIP3 6kyr BP simulations with the IPSL model (Kageyama et al. 2013) we
decided to only consider natural aerosols. In the IPSL model, aerosols are accounted for by prescribing the
optical distribution of dust, sea-salt, sulfate and particulate organic matter (POM), so as to take into account the
aerosol forcing in the radiative code (Dufresne et al., 2013). In PMIP3 simulations these variables where
prescribed to 1860 CE values, which correspond to the beginning of the industrial area for which the level or
sulfate and POM is slightly higher than the values found in the Holocene (Kageyama et al., 2013a). Here we
prescribe only dust and sea-salt and neglect the other aerosols. This choice was driven by the fact that we also
plan to run simulations with fully interactive dust and sea-salt.



Most of the tests done to set up the model version follow the PMIP3 protocol (Braconnot et al., 2012).

But the transient simulation, and thus the long mid Holocene simulations used as initial state for it, both follow

the PMIP4-CMIP6 protocol (Otto-Bliesner et al., 2017, Tab. 1). For PMIP4-CMIP6 simulations, the latest

estimate of trace gazes ($CO_2$, $CH_4$ and $N_2O$) from ice cores are imposed as boundary conditions, to have a

consistent history of the evolution of these gazes across the Holocene (Otto-Bliesner et al., 2017). We run a 1000

159        year-long simulation to produce 6ka BP initial conditions in equilibrium with the external forcing (insolation,

trace gazes and aerosols) that can be used as initial state for the transient late Holocene simulations. The version

with interactive vegetation needs also to be integrated long enough to build the vegetation cover in equilibrium

with the mid-Holocene climate (see section 3).

**2.3     Impact of model version and forcing strategy on mid-Holocene climate**

Figures 1a and b compare the results of a MH simulation using the new hydrology and snow model

when forced with PMIP4 boundary conditions to the PMIP3-CMIP5 MH simulation (MH-FPMIP4) with the

standard IPSLCM5A-LR version of the IPSL model (MH-PMIP3). The simulated MH climate is globally

warmer in MH-FPMIP4, except over tropical forests in Africa and Amazonia, and in East Asia and Siberia (Fig.

1b). It is associated with larger precipitations in the tropics and in mid latitudes, and with reduced precipitation

in the subtropics (Fig. 1a). These differences in MH climatology between the two simulations result from both

the changes in the configuration of the land surface model and the changes in forcing. Table 1 presents the major

simulations done to test some of the last model improvements and tuning that affect the global energy and

hydrological cycles. They all keep exactly the same set of adjusted parameters as in Dufresne et al. (2013) for the

ocean-atmosphere system. The additional adjustments only concerned the land surface model and the forcing

factors.

All the simulations were run long enough (300-1000 years) to reach a radiative equilibrium and be

representative of stabilized MH climate (Fig. 2). They are free of any artificial long term trends after the

adjustment phase and the global averages of the surface flux and the radiative budget at top of the atmosphere

close are close to zero (i.e. 0.4 $W.m^{-2}$). This closure of the surface fluxes is equivalent to the one in previous

IPSL PMIP3 MH simulation (Kageyama et al., 2013a). Figure 2 also highlights that the new hydrological model

(L11) produces about 1.25 $mm.d^{-1}$ higher global annual mean evaporative rates than MH PMIP3, but that this

higher evaporation is achieved with similar global mean temperature. The water cycle is more active in L11. It

has implications on the geographical distribution of precipitation and temperature compared to the MH PMIP3

simulations (Fig. 1c). With the new hydrology, precipitation is enhanced in the mid-latitudes and over the

tropical lands where larger evapotranspiration and cloud cover both contribute to cool the land surface (Fig. 1d).

Part of the land surface cooling as due to a high fresh snow albedo in this first L11 version of the land surface

model. In the tropical region, the Amazon basin is more humid, as is the Indian monsoon. West Africa is slightly

less humid, whereas precipitation is increased in equatorial Africa and over the Gulf of Guinea (Fig. 1c).

Similarly, precipitation is increased in the western part of the Indian Ocean and decreased over the maritime

continent and along the equator in the Pacific Ocean (Fig. 1c). Interestingly, the cooling over land is

compensated at the global scale by a warmer surface ocean (Fig. 1d).

The changes in the way aerosols are considered in the transient simulations have an impact on the

global model adjustment. Only considering dust and sea salts lead to a radiative difference of about 2.5 $W.m^{-2}$ in



external climate forcing compared to previous simulations, as seen by the heat budget imbalance at the surface at
the beginning of the L11Aer simulation (Figure 2). When it is implemented in the coupled model simulations
this additional forcing leads to excess energy at the surface and an increase of the 2m air temperature. The global
scale adjustment of the model is achieved in approximately 250 years when the surface heat budget becomes
close to 0 (Fig. 2a), but global air temperature has increased by 1.5 °C. The largest warming over land is found in
the northern hemisphere, but the ocean warms almost everywhere, except in the Antarctic circumpolar current,
by about 1°C (Fig. 1f). In the southern hemisphere the subduction of surface waters and insulator effect of sea-
ice explain that the surface remains cooler than in the other regions (Fig. 2f). These warmer conditions favors
higher precipitation over the tropical ocean and in mid-latitude with a global pattern rather similar to what is
expected in simulation of global warming induced by increased atmospheric $CO_2$ (Fig. 2e). Note that a similar
offset in external forcing is also present in the pre-industrial simulation in this case. The effect on the differences
between mid-Holocene and pre-industrial climate might be small compared to the effect on mean climatology for
a given period.

In figure 1 the larger precipitation in L11 compared to PMIP3 can be partially explained by larger

evaporation resulting from higher evaporation rate of bare soil, which appeared to be too high in intermediate
seasons. The model bare soil evaporation is exacerbated by the fact that the way the mosaic vegetation is
constructed favors too much bare soil when leaf area index (LAI) is low (Guimberteau et al., 2018). To
overcome this problem, an artificial 0.70 factor was implemented to limit bare soil evaporation (Table. 1). All
the other surface type remains as they are in L11. This factor is compatible with the order of magnitude of the
reduction brought by the implementation of a new evaporation parametrization for bare soil in later IPSLCM6A
version of the model (Peylin et al. pers. com.). The second one concerns the combination of snow albedo with
the vegetation albedo. The procedure was different when vegetation was interactive or prescribed. In both cases
the albedo results now from a combination of snow and vegetation albedo based on the effective vegetation
cover in the grid box, which put a substantial weight on bare soil albedo when LAI is small. The albedo becomes
thus larger in simulations in which the vegetation is prescribed compared to the IPSL-CM5A-LR reference
version of the model. It counteracts the effect of the fresh snow albedo reduction.

Since we are dealing with a coupled system, some of these changes didn't lead to the direct expected

changes on the model climatology due to internal feedbacks in the coupled system. In particular, the reduction of
bare soil evaporation didn't reduced evaporation as expected. This is due to the temperature feedback in the
coupled system. Indeed, when evaporation is reduced, soil temperature increases and the regional climate get
warmer allowing for more moisture in the atmosphere and thereby more evaporation where soil can supply water
(Figure 1 g and h and Fig. 2). Therefore, the difference does not show up on the precipitation map (Fig. 1g) but
on the increased temperature over land in the northern hemisphere (Fig. 1h). It is consistent with similar findings
when analyzing land use feedback (Boisier et al., 2012). In our case, it partly counteracts a model cold bias in
these regions. This unexpected results with a forced vegetation model reasoning stresses once that fast feedbacks
occur in coupled systems and that any comparison of surface fluxes should consider both the flux itself and the
climate or atmospheric variables used to compute it (Torres et al., 2018). Note that in figure 1h the small global
warming is still a footprint of the warming induced by the aerosol effect described above.

Finally, compared to the 11LAerEV simulation the cooling found for the MH-FPMIP4 simulation

reflects the difference between the PMIP3 and PMIP4 external forcing. The difference in forcing was estimated





to -0.8 W.m$^{-2}$ by Otto-Bliesner et al. (2017). This is the order of magnitude found for the imbalance in surface
net surface heat flux at the beginning of the MH-FPMIP4 simulation that started from L11Aer run with PMIP3
protocol (Fig. 2a). As expected, it leads to a slight cooling and corresponding reduction of evaporation and
precipitation.
**2.4    How good is this version compared to present day climatology?**

The way the different changes affect the model climatology is similar for the mid-Holocene and

preindustrial climates. We only run a pre-industrial simulation PI with the version including all changes (PI-
FPMIP4, Tab. 1). This allows us to objectively assess if the introduction of the new hydrology and the
adjustments degrade or improve the model results compared to the IPSLCM5A-LR CMIP5 simulation (PI-
PMIP3, Tab. 1).

A rapid overview of model performances is provided by a simple set of metrics derived from the metric

package (Gleckler et al., 2016), where the new version is compared to PI-PMIP3 and to all the other available
CMIP5 PI simulations (Fig. 3). This figure highlights that the annual mean model bias is reduced for
temperature, at about all model levels but enhanced for precipitation and total precipitable water (Fig. 3a). This
echoes the analyses above showing that precipitation is increased in the 11 layer soil hydrology due to larger
evaporation. The evaporation and precipitation biases are reinforced by the warming induced by the offset in
radiative forcing we introduce by only considering dust and sea-salt aerosols. The latter however also contributes
to reduce temperature biases. Despite this precipitation bias that slightly degrades the overall model
performances compared to the CMIP5 ensembles (Fig. 3b) the model performs quite well compared to the other
CMIP5 simulations, except for cloud radiative effect. The effect of cloud in the IPSLCM5A-LR simulations has
already been pointed out in several manuscripts and results mainly from low level clouds over the ocean
(Braconnot and Kageyama, 2015; Vial et al., 2013). Note that the atmospheric tuning is exactly the same as in
the default IPSLCM5A-LR version, and that the changes described above have almost no effect on the cloud
radiative effect. Overall the model version with the 11 layers hydrology has similar skill as the IPSLCM5A
reference (Dufresne et al., 2013) and we are confident that the version is sufficiently realistic to serve as a basis
on top of which we can include the dynamical vegetation.
**3    Mid-Holocene simulations with interactive vegetation**
**3.1    Initialization of the mid-Holocene dynamical vegetation and simulated mid Holocene climate**

Two different strategies have been tested to initialize the dynamical vegetation (Table 2). In the first one

(Vmap), the vegetation distribution was obtained from an off line simulation with the land surface model that has
been forced by CRU-NECP 1901-19010 climatology (Viovy, 2018) regrided on the IPSLCM5A-MR model
resolution. The resulting map was then prescribed as initial state in the coupled model and the dynamical
vegetation was switched on to run a mid-Holocene simulation (Fig. 4). In the second case (Vnone), the model
restarted from bare soil with the dynamical vegetation switched on, using the same initial state as for the
previous simulation for the atmosphere, the ocean, sea-ice and land-ice. Despite a tendency to converge to
different solutions in the beginning of the simulation (black and blue curves in Fig. 4 a,b, and c), the two
simulations converge with very similar global vegetation cover over a longer time scale after that the PMIP4



instead of PMIP3 mid Holocene boundary conditions were applied to the model (red and yellow curves in Fig. 4
a,b, and c). It suggests that there is only one global mean stable state for the mid-Holocene with the IPSL model,
irrespective of the initial vegetation distribution.

Compared to the reference vegetation used when vegetation is prescribed to modern values (green line

in Fig. 4 d, e, and f), the bare soil cover is reduced and grasses and trees occupy a larger land fraction (Fib. 4 b
and c). Note however that the global averages mask small differences in regional vegetation cover (Figure 5 a, d,
and g). MH Vmap reproduces slightly more trees in West Africa and less trees north of 60°N than Vnone (Fig.
5g). Over most of these grid points the differences in trees are compensated by grass (Fig. 5d) except to the south
of the Tibetan plateau where bare soil is dominant in Vmap (Fig. 5a).

Figures 6a and b indicate that the simulated MH climate with interactive vegetation is warmer than the

simulation with prescribed vegetation over the continents and in the South Atlantic Ocean. It also highlights that
precipitation is increased over the African tropical forest and reduced over South America. Over Eurasia, part of
the warming comes from the fact that there is cropland in the 1860 CE vegetation map when vegetation is
prescribed (Fig. 3). When the dynamical vegetation is active, the resulting map only includes natural vegetation.
In most of Eurasia forest replaces croplands (Fig. 6f). The lower forest albedo induces warmer surface conditions
in these regions. Also when snow combines with forest instead of grasses, the snow/vegetation albedo is lower
leading to the positive snow-forest feedback widely discussed for the last glacial inception (de Noblet et al.,
1996; Kutzbach et al., 1996). The plus minus features over the tropical ocean suggest a slight shift in the location
of the Inter Tropical Convergence Zone (ITCZ) and South Pacific Convergence Zone (SPCZ), whereas over
South America it mainly shows reduced precipitation in the west and a slight increase in the east (Fig. 6a). These
large scale patterns result from large scale changes in atmospheric and ocean circulations induced by differences
in the land-sea contrast and regional changes in vegetation.
**3.2     Simulated versus reconstructed mid-Holocene vegetation**

The vegetation dynamics module simulates fractional cover of each PFT, which cannot be directly

compared with the reconstruct biome types based on pollen and plant macrofossil data from the BIOME 6000
dataset (Harrison, 2017). In order to facilitate the comparison, we use a biomization method to convert modeled
vegetation properties into the eight "megabiomes" provided by BIOME 6000 (Fig. 7). The algorithm, uses a
mixture of simulated climate and vegetation characteristics (see Fig. A2,). The default values for each threshold
are the same as in Zhu et al. (2018). Several sensitivity tests with alternative thresholds proposed in previous
studies (Joos et al., 2004; Prentice et al., 2011) have been done to account for the uncertainties in the biomization
methodology (see Fig. A2). They provide similar results as the one provided for PI-VNone in figure 7. It also
shows that, as expected from figure 5, Vnone and Vmap produce very similar results.

At first look PI-Vnone reproduces the large scale pattern found in the BIOME6000 (Fig. 7a). The

comparison however indicates that the boreal forest tree line is located too far south, which suggests a cold bias
in temperature in these regions. Also vegetation is underestimated in West Africa, consistent with a dry bias (not
shown). The underestimation of the African monsoon precipitation is present in several simulations with the
IPSL model (Braconnot and Kageyama, 2015), and is slightly enhanced in summer when the dynamical
vegetation is active. With interactive vegetation however equatorial Africa is more humid (Fig. 6a).





Figure 7c provides an idea of the major mismatches between simulated vegetation and the

reconstructions. A perfect match with the biome reconstruction would only produce values on the diagonal. The

overall percent of correctness at the reconstruction sites is about 50%. In particular the simulation produces too

much desert where we should find grass and shrub. It also produces too much tundra instead of boreal forest, and

too much Savanah and dry woodland in several places that should be covered by temperate-tree, boreal-tree or

tundra, confirming the visual map comparison (Fig. 7c).

### 3.3    Comparison with the pre-industrial climate

We also tested the results of the dynamical vegetation in simulations of the preindustrial climate (dark

pink and orange lines in Fig. 4d, e and f), to check if PI vegetation and climate would also be similar when

starting from MH-Vmap or MH-Vnone. This is also a way to have a better idea of the range of response one

would expect from ensemble simulations, knowing that we will only run one full transient simulation with

interactive vegetation. Simulated climate and vegetation biases also impact the representation of the vegetation

cover when vegetation is fully interactive in the model. They also need to be accounted for to assess the response

of vegetation to insolation forcing.

For the PI-Vmap simulation, the orbital parameters and trace gazes were first prescribed to pre-

industrial conditions for 15 years while the vegetation map allocating the different PFTs in each grid cell was

prescribed to the vegetation map obtained in MH-Vmap (Tab. 2, Fig. 4). Then, the dynamical vegetation was

switched on. Since surface variables adjust rapidly, this is a way to compare the rapid adjustment to insolation

and the additional effect due to the dynamical vegetation (not discussed here). The switch to dynamical

vegetation induces a rapid transition of the major PFTs that takes about 10 years before a new global equilibrium

is reached (Fig. 4 d, e and f). For PI-VNone the same procedure was applied, but the dynamical vegetation was

switched on after 5 years (Tab. 2 and Fig. 4). For this simulation, vegetation converges rapidly to the new

equilibrium state, without any relaxation or rapid transition.

PI-Vnone and PI-Vmap converge to different global vegetation states (Fig. 4). In particular PI-Vmap

produces a larger bare soil cover than PI-Vnone (Fig. 4 d). It is even larger than the total bare soil cover found in

the 1860 CE map used in PI simulations when vegetation is prescribed (Fig. 4). Interestingly part of these

differences between Vmap and Vnone, are found in the southern hemisphere and the northern edge of the

African and Indian monsoon regions (Fig. 5b). The differences in the tree cover in the northern hemisphere is

also slightly enhanced compared to the one found between these two simulations for the corresponding MH

simulations (Fig. 5). These differences in PI vegetation explain the vegetation differences between MH and PI

(Fig. 8). The simulated changes seem larger with Vmap. Previous assessment of model results against vegetation

and paleoclimate reconstructions (e.g. Harrison et al., 2014; Harrison et al., 1998) suggest that MH – PI

vegetation for Vmap would look in better agreement with reconstructed changes from observations in terms of

forest expansion in the northern hemisphere or grasses in Sahel (Fig. 7 c, d, e and f). However the modern

vegetation map for this PI-Vmap simulation has even less forest than PI-Vnone north of 55°N (Fig. 4 e, f and i),

for which forest is already underestimated (not shown). These differences in PI vegetation have only a small

counterpart in climate. It corresponds to cooler condition in the mid and high norther latitude (Fig. 6f). In annual

mean there is almost no impact on precipitation (Fig. 6e).



Compared to the version with the 11 layer hydrology (PI-FPMIP4) both PI-Vmap and PI-Vnone have
larger temperature biases, mainly because of the Northern (NH ) hemisphere warming induced by vegetation
(Fig. 6b). It brings the global performances for temperature close to the IPSLCM5A-LR CMIP5 version. It also
contributes to reduce the mean bias in precipitable water, evaporation, precipitation and long wave radiation. It
has no effect however on the bias pattern (assessed by the rmst in Fig. 4). Figure A1 (see annex) further shows
that the performances of PI-Vnone and PI-Vmap are very similar, and closer to each other than to other
simulations, whatever the season or the latitudinal band. The small differences in climate listed above are thus
too small to be captured by global metrics. It suggests that there is not direct relationship between the different
vegetation maps and model performances. The different vegetation maps are obtained with a similar climate,
which indicates that in this model multiple global and vegetation states are possible under pre-industrial climate
or that tiny climate differences can lead to different vegetation cover in the northern hemisphere. Results for the
southern hemisphere are more puzzling.
**4    Simulated climate and vegetation throughout the mid to late Holocene**
**4.1    Initial state and experimental design for the transient simulations.**

Previous section indicates that there are very little differences in terms of climate between PI-Vnone
and PI-Vmap, but that the simulated vegetation for the PI climate is substantially different. In particular PI-
Vnone produces less bare soil and more forest in mid and high northern latitudes (Fig. 5). The major drawback is
that West Africa is slightly less satisfactorily represented in PI-Vnone simulation. Despite this bias, we decided
from a global perspective to use a 1$^{st}$ of January obtained after 500 year in MH-Vnone-FPMIP4 as initial state
for the transient TRHOLV simulation (Tab. 2).

For this simulation the trace gazes vary every year using one of the latest reconstructions for $CO_2$, $CH_4$
and $N_2O$, that has been provided by Joos (see Otto-Bliesner et al., 2017). The atmospheric $CO_2$ concentration is
slowly rising throughout the Holocene from 264 ppm 6000 years ago to 280 for the pre-industrial climate around
-100 PB (1850 CE) and then experiences a rapid increase from -100 BP to 0 BP (1950 CE) (Fig. 9). The methane
curve shows a slight decrease and then follows the same evolution as $CO_2$, whereas $NO_2$ is almost flat
throughout the period. The impact of the small variations in atmospheric trace gazes is small over most of the
Holocene (Joos and Spahni, 2008). The largest changes in these trace gazes occurred with the industrial
revolution, so that they have an imprint of about 1.28 W.m$^{-2}$ additional forcing in the atmosphere compared to
MH, most of which occurs in the last 100 years.

The major forcing comes from the slow variations of the Earth's orbital parameters. The change in
seasonality is the dominant factor that affects climate variations over most of the Holocene, except in the last
part of the simulations from 2000 years BP onward (Fig. 10). The changes in seasonality correspond to decrease
seasonality in the northern Hemisphere and increased seasonality in the southern Hemisphere. Note however that
the timing of the changes for the different seasons (Winter, NDJF, i.e. November to February average, and
Summer, JJA, June to September average) is slightly different between the hemispheres, which modulates the
interhemispheric contrast with time.





### 4.2 Long term climatic and vegetation trends


Changes in temperature and precipitation follow the long term insolation changes in each hemisphere
and for the different seasons until about 2000 yrs BP to 1500 yrs BP (Fig. 10). Then trace gazes and insolation
forcing become equivalent in magnitude and small compared to MH insolation, until the last period where trace
gazes lead to a rapid warming in both hemispheres. The NH summer cooling reaches about 0.8 °C and is
achieved in 4000 years. The last 100 year warming reaches 0.6 °C and almost counteracts, for this hemisphere
and season, the insolation cooling. SH summer and NH Winter conditions (NDJF) are both characterized by a
first 2000 years warming induced by insolation. It reaches about 0.4°C. It is followed by a plateau of about 3000
years before the last rapid increase of about 0.6°C that reinforces the effect of the Holocene insolation forcing.
During SH winter temperature does not seem to be driven by the insolation forcing (Fig. 10 d). In this
hemisphere part of the insolation forcing is absorbed in the ocean (not shown), which dampens the surface
temperature warming. In both hemispheres precipitation trends are well correlated to temperature trends, as it is
expected from a hemispheric first order response driven by Clausius Clapeyron relationship (Held and Soden,
2006). This is not the case for winter conditions because one needs to take into account the changes in the large
scale circulation that redistribute heat and energy between regions and hemispheres (Braconnot et al., 1997;
Saint-Lu et al., 2016).
Interestingly temperature and precipitation exhibit centennial variability that is not present in the
imposed insolation and trace gazes forcing. It is the results of all the internal interactions between the physical
climate, carbon and dynamical vegetation. Because of this it is difficult for example to say if the NH hemisphere
winter temperature trend was rapid until 4000 years BP and then temperature remains stable, or if the event
impacting temperature and precipitation around 4800 to 4500 BP masks a more gradual increase until 3000 BP
as it is the case for NH Summer where the magnitude of the temperature trend is larger than variability (Fig. 10
b). Note that some of these internal fluctuations reach half of the total amplitude of the trend, even with the 100
year smoothing applied before plotting. Temperature and precipitation are well correlated at this centennial time
scale and hemispheric scales for all seasons.
The associated vegetation trends correspond to reductions or increases reaching 2 to 4% of total land
areas depending on vegetation type. It is consistent with the order of magnitude found in figure 4 between the
MH and PI simulations (Fig. 11). It follows the insolation forcing trend in both hemispheres. It is thus opposite
in the two hemisphere, except for the last part where the recent period reflects the rapid increase of atmospheric
$CO_2$ concentration. In addition this long term evolution parallels the evolution of temperature and precipitation,
with a good correlation with summer conditions (Fig. 10). As expected, the global vegetation averages reflect the
northern hemisphere changes where most of the vegetated continental masses are located.  The largest trends are
found for tree and grass covers in both hemispheres, with the exception of the last 100 year period where bare
soil variation are relatively larger than for the whole mid to late Holocene. The gross primary productivity (GPP,
Fig. 11 d) is driven in both hemispheres by the changes in tree cover. It accounts for a reduction of about 5
$PgCy^{-1}$. The GPP increase in the last 100 years results from increased atmospheric $CO_2$. It is however possible
that the GPP change is underestimated in this simulation because $CO_2$ is prescribed in the atmosphere, which
implies that the carbon cycle is not fully interactive.



### 4.3 Regional trends


Figure 12 highlights relative differences for three regions that respectively represent climate conditions

north of 60°N, over the Eurasian continent, and in the West African monsoon Sahel/Sahara region. These are
regions for which there are large differences in MH – PI climate and vegetation cover (Fig. 6 and 8). They have
also been chosen because they are widely discussed in the literature and are also considered as tipping points for
future climate change (Lenton et al., 2008). A complete evaluation of the simulated trends and timing of the
changes is out of the scope of this paper. However, these regions are well suited to provide an idea of different
characteristics between regions.

North of 60°N and in Eurasia a substantial reduction of tree at the expense of grass starts at 5000 years

BP (Fig. 11). Vegetation has almost its pre-industrial conditions around 2500 years BP. Interestingly the largest
trends are found between 5000 years BP and 2500 years BP in this region and this reflects well the timing of the
NH hemispheres summer cooling. The change in total forest in Eurasia is small. A first step change is followed
by a second one around 3000 years BP. The NH decrease in forest cover is mainly driven by the changes that
occur north of 60° N (Fig. 11, 12 and 14 g). Despite the vegetation biases in high latitudes discussed in section 3,
these trends reflects more or less what is expected from observations (Bigelow et al., 2003; Jansen et al., 2007;
Wanner et al., 2008). Even though the curves are smoothed by a 100 years average, they exhibit substantial
centennial variability north of 60°N and in Eurasia (Fig. 12 a and b). The magnitude of this variability represents
up to half of the total signal north of 60°N and up to the maximum change in Eurasia. Over West Africa (Fig. 12
c), and the largest trends starts slightly later (4500-5000 years BP) and are more  gradual until 500 years BP. The
vegetation trends are also punctuated by several centennial events that do not alter much the long term evolution
as some of these events do in the other two boxes. The reduction in forest for grasses north of 60°N and increase
in bare soil at the expense of grasses in West Africa lead to reduced GPP (Fig. 13), except for the last part in
high latitudes when tree cover regrows when $CO_2$ increases. This effect is consistent with the observed historical
growth in gross primary production discussed by Campbell et al.  (2017).

Figure 12 provides the feeling that there are only marginal changes in Eurasia in terms of vegetation. It

is partially due to the fact that the total tree cover does not reflect well the mosaic vegetation and forest
composition.  Figure 15 shows the relative change of the different types of forest found over Eurasia. It shows
that the long term decrease in forest is dominated by the decrease in temperate and boreal deciduous trees.
Boreal needleleaf evergreen trees do not change whereas the temperate ones increase. The different trees have
also different timing and variability. This figure highlights that the long term change in Eurasian tree
composition throughout the mid to late Holocene is also punctuated by centennial variability. The large events
have a climatic counterpart (Fig. 10), so that the composition of the vegetation is certainly the results of a
responses to the long term climatic change and to variability that can lead to different vegetation composition
depending on stable or unstable vegetation states (Scheffer et al., 2012).  Rapid changes and variability has been
discussed for recent climate  in these regions (Abis and Brovkin, 2017), which suggests that despite the fact that
our dynamical vegetation model might underestimate vegetation resilience the rapid changes in vegetation
mosaic induced by long term climatic trend and variability in this transient simulation deserve attention.



### 4.4 The PI and historical period in the transient simulation

Sever studies suggest that the initial state has only minor impact on the final climate because there is
almost no changes in the thermohaline circulation over this period and models do not exhibit major climate
bifurcations (e.g. Bathiany et al., 2012). This is the main argument used by Singarayer et al. (2010) to justify that
their suite of snap shot experiments may provide reasonable transient climate vision when put together. It is the
case in the TRHOLV simulation when vegetation is fully interactive? This transient simulation does not exhibit
much change in indices of thermohaline circulation that remains close to 16-18 Sv (1 Sv = $10^6 m^3.s^{-1}$) throughout
the period. The preindustrial climate (1860 CE) corresponds to the climate around 100 BP in the TRHOLV
simulation (Fig. 10). The global metrics (Fig. 3) show that at the global scales the results of the TRHOLV
simulations are similar to those of PI-Vnone. It is also the case for seasonal and extratopical/tropical values (Fig.
A1). We can therefore conclude that there is no difference in mean surface climate characteristics between the
snap shot PI-Vnone experiments and the PI period simulated in transient TRHOLV simulation.
Then, is the vegetation also similar to the one simulated in PI-VNone? The MH minus PI differences
and the PI vegetation and simulated in TRHOLV (Fig. 14 a, d, and g, and c, f, and i) shows little differences to
the one found for PI-Vnone (Fig. 8 b, d, and f, and Fig. 14 b,e,and h). The relative percentages of land covered
by the different vegetation classes correspond to 15% for bare soil, 41% for grass and 43% for tree respectively.
These values are similar to the one found for PI-VNone (15%, 40% and 44% respectively) within 1% error bar.
They are both different from those of PI-Vmap (20%, 37% and 43%). It suggests that the adjustment time is long
enough to converge to similar solutions. It thereby questions why we found different PI climate-vegetation state
between PI-Vmap and PI-Vnone. This doesn't necessarily hold at the regional scale where regional differences
are also found between PI-THROLV and PI-Vnone. Indeed, Figure 15 b, e, and h indicate differences in tree and
grass cover in Eurasia around 60°N and different geographical coverage between bare soil, grass and trees over
South Africa and Australia. Further investigation would be needed to fully assess these differences and analyze
the possible role of variability in these differences.
The last point to mention is the fact that the effect of trace gazes and in particular of the rapid increase
of the atmospheric $CO_2$ concentration over the last part of the simulation has also a strong impact on the
evolution of the natural vegetation. When reaching 0k BP (1950 CE), bare soil remains close to PI, grass reduces
by 3% and tree increases by about 3%. Interestingly this tree recovery counteracts the reduction from mid
Holocene in mid and high NH latitudes (Fig. 15 f). Bare soil is only slightly higher and grass smaller. It is not
possible here to properly assess the historical climate and vegetation cover of THROLV. In the real world, they
have been both affected by land-use that is neglected here. Nevertheless, our results raises once more that for
model data comparison, the reference period is of great importance to be able to fully assess model results. They
also remind us that the historical period is unusual in the context of the mid to late Holocene.

### 5 Conclusion

This long transient simulation over the last 6000 years with the IPSL climate model is still one of the
first simulations over this period with a general circulation model to include a full interactive carbon cycle and
dynamical vegetation. We show that, despite some model biases that are amplified by the additional degree of
freedom resulting from the coupling between vegetation and climate, the model reproduce reasonably well the





large scale feature expected from the observation over this period. There has been lots of discussion on the sign
of the trends in the northern mid-latitude following the results of the first coupled ocean-atmosphere simulation
with the CCSM3 model across the deglaciation. Our results seem in broad agreement with the 6000 to 0 part of
the revised estimates by Marsicek et al. (Marsicek et al., 2018). There is little change in annual mean throughout
the last 6000 years (not shown). The seasonal cycle is the main driver of the climate and vegetation changes.

Several points emerge from this study. The first one is that the long term evolution of vegetation cannot
be characterized by a linear trend from the mid-Holocene to the preindustrial climate. The major changes occur
between 5000 and 2000 year BP and the exact timing depends on regions. In our simulation the forest reduction
in the northern hemisphere starts earlier than the vegetation changes in Africa. It also ends earlier. The last
period, starting about 2000 years ago reflects the increase in trace gases with a rapid regrowth of tree in the last
100 years when $CO_2$ and temperature increase at a rate not seen over the last 6000 to 2000 years. Some of these
results already appear in previous simulations with intermediate complexity models (Crucifix et al., 2002;
Renssen et al., 2012). Using the more sophisticated model with a representation of different types of tree brings
the new results that even though the total forest cover does not vary much throughout the Holocene in TRHOLV,
the composition of the forest varies more substantially, with different relative timing between the different PFTs.
The analysis of the linkage with long term climate trends, variability and internal vegetation instability would
require further investigation. It would guide the development of methodologies to assess the vegetation
instabilities in this region seen in the recent period (Abis and Brovkin, 2017), as well as the discussion on the
internal instability of vegetation that could be partly driven by climate noise (Alexandrov et al., 2018). I might
also be an important aspect to consider for future model data-comparison.

As discussed in section 3 and 4, the vegetation differences between PI-Vmap and Pi-VNone raise once
more the possibility for multiple vegetation equilibrium under pre-industrial or modern conditions as it has been
widely discussed previously (e.g. Brovkin et al., 2002; Claussen, 2009). Here we have both global and regional
differences. Our results is however puzzling, because we only find limited differences between the PI-Vnone
snapshot simulation and the PI climate and vegetation produced at the end of TRHOLV. These simulations start
from the same initial state and in one case PI condition are switch on in the forcing, whereas the other case the
6000 years long term forcing in insolation and trace gases is applied to the model. An ensemble of simulations
would be needed to fully assess vegetation stability. In the northern hemisphere and over forest areas, MH-Vmap
produced slightly less trees that MH-Vnone. It might have been amplified by snow albedo feedback under the PI
conditions that are characterized by a colder than MH climate in high latitudes in response to reduced incoming
solar radiation associated with lower obliquity. The differences between the southern and northern hemisphere
characterized by large differences in grasses and bare soil are more difficult to understand and suggest different
response to the changes in southern hemisphere seasonality. This is in favor of different equilibrium induced
only partly by climate-vegetation feedback. We need also to raise the point that part of these differences could
also be due to internal modeling and full consistency between the imposed and dynamical part of the system.
However these would not explain why vegetation is sensitive to initial state in PI and not in MH. We would
expect that similar differences would be found in that case between the two periods. It is also possible that the
climate instability induced by the change from one year to the other in insolation and trace gazes lead to rapid
amplification of climate in high latitude and that vegetation in the southern hemisphere move from one instable
state to the over. The strongest conclusion from these simulations is that the vegetation-climate system is more





sensitive under the pre-industrial conditions (at least in the northern hemisphere latitudes).  In depth analyses of
the fast vegetation response and of its linkages/or not with interannual to multi-decadal variability is needed. The
different time scales involved in this long term evolution can be seen as an interesting laboratory for further
investigation in this respect.

In this study we also points on the difficulties to fully assess model results. The reason is that we only
represent natural vegetation, and neglect land use and also aerosols other than dust and sea-salt. Therefore the PI
and historical climate cannot be realistically reproduced, even though most of the characteristics we report are
compatible with what has been observed. It also clearly shows that assessment of the magnitude of the simulated
differences between MH and modern conditions depends on the reference period. This has implication for
model-data comparisons, but also for reconstruction of temperature or moisture from paleoclimate archives that
are in general calibrated using specific datasets. Similar methodologies for data sampling need thus to be applied
both on paleoclimate records and on model outputs. It also suggest that more needs to be done to assess the
processes leading to the observed changes rather than the changes themselves.

Since the MH-PI changes in climate and vegetation is similar in our simulation between snapshot
experiments and a long transient simulation we can wonder what we learn out of this long simulation. What is
the value added of a transient versus a snapshot experiment if climate differences are similar?  Here also we do
not have definitive answers. The good point is that model evaluation can be done on snapshot experiments,
which fully validate the view that the mid-Holocene is a good period for model benchmarking in the
Paleoclimate Modeling Intercomparison Project (Kageyama et al., 2018). However the MH – PI climate
conditions mask the long term history and the relative timing of the changes. We also mainly consider here
surface variables that have a rapid adjustment with the external forcing. In depth analyses of ice covered regions
and of the ocean response would be needed to assess if this is valid for all the aspects of the climate system. Also
we only consider long term trends in this study, but it shows that centennial variability plays an important role to
shape the response of climate and vegetation to the Holocene external forcing at regional scale. Lots of changes
can also be reported on interannual to multidecadal variability that would require further investigation. For these
time scales further investigation is needed to tell if the characteristics of variability depends or not on the pace of
climate change.

## 6    Annex

### 6.1    A1 Spatio-temporal agreement between model results and observations in the extratropics and tropics

Figure 3 highlights the model-observation agreement for the pre-industrial climate considering global
metrics (Gleckler et al., 2016; Gleckler et al., 2008). Even though these metrics take into account the simulated
patterns, it is possible that they do not capture well differences between model versions and between model and
observations over part of the globe. We therefore complete the analyses by computing the same metrics (bias and
root mean square) at the seasonal time scale and for 3 latitudinal bands. We restrict the figure to surface air
temperature and precipitation that reflects well the differences. It shows that these measures capture differences
between the IPSLCM4A-LR version of the IPSL model (Dufresne et al., 2013) and the new version developed



for the TRHOLV transient simulation (see section 2). It also highlights the impact of running the model with the
dynamical vegetation. However, as in Figure 3 the simulations with different MH conditions for the interactive
vegetation, as well as the PI conditions obtained after 5900 years of transient simulation are difficult to
distinguish. Differences become significant again when considering the last 50 years of the transient simulations
that are affected by increase greenhouse gases.
**6.2    A2 Biomization and sensitivity analysis.**

To convert the ORCHIDEE model PFTs into mega BIOMES we use the same algorithm than Zhu et al.

(2018). Figure A1a shows the different thresholds used in the algorithm. The black numbers correspond to the
default values used to produce Figure 6 in the main text. Since some of these thresholds are somehow artificially
defined, we also tested the robustness of our comparison by running sensitivity tests. These test considered
successively different threshold in Growing Degree Days above 5°C (GDD5), canopy height and foliage
projective cover as indicated in red on figure A1a.

The different thresholds induce only slight difference on the BIOME map for a given simulation. The

largest sensitivity is obtained for the height. When 10 m is used instead of 6 m, a larger cover of savannah and
dry woodland is estimated from the simulations in mid and high norther latitudes.  In these latitudes also a large
sensitivity is found when the GDD5 limit is set to 500 °C. d$^{-1}$ instead of 350 °C.d$^{-1}$ between tundra and savanah
and dry woodland or boreal forest.

The same analyses transformation into megabiomes was performed for the Vmap and Vnone

simulations. Similar sensitivity is found to the different thresholds for these two simulations (figure A1b). The
comparison of the different maps show that, as already stated from Figure 5, small differences can be found in
the vegetation distribution, mainly on the forest cover in mid and high latitude. The synthesis of the goodness of
fit between model and data in figure A1c. It shows that the two simulations provide as expected very similar
results when compared to the BIOME6000 map. It is interesting to note that the different thresholds do not have
a large impact on the model data comparison. The change in GDD5 limit produces tundra in better agreement
with pollen data, and the canopy height better results with savannah and dry woodland. Note however that this
result is in part due to the fact that there is little data in regions where the impact is the largest (Figure 6 in the
main text).

*Acknowledgments.* We would like to thanks our colleagues from the IPSL global climate model group
for their help in setting up this intermediate version of the IPSL model. In particular the ORCHIDEE group
provided good advices for the closure of the hydrological cycle in the land surface scheme (Philippe Peylin,
Agnès Ducharne, Fréderic Cheruy and Joséfine Gattas) or the snow ablation (Sylvie Charbit and Christophe
Dumas). The workflow for these long simulations benefits from the development of Anne Cozic and Arnaud
Caubel. Discussions with Philippe Ciais and Yves Balkansky were also at the origin of the choice of the land
surface model complexity and aerosols forcing strategy. Pascale Braconnot and Olivier Marti have been awarded
a PRACE computing allocation (THROL project) to start the simulations, as well as a GENCI specific high end
computing allocation and normal allocation time (gen2212). This work is supported by the JPI-Belmont
PACMEDY project (N ° **ANR-15-JCLI-0003-01).**

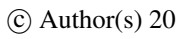




| Simulation | Comment | Initial state |
|---|---|---|
| **midHolocene (MH)** | | |
| MH_PMIP3 * | Reference PMIP3-CMIP5 IPSL simulation (Kageyama et al., 2013a) | Previous MH long term simulation with the model used to test model configuration |
| MH_L11 (S_Sr01) | As PMIP3, but with new version of land surface model (hydrology and snow model | From the last MH test of the new model configuration (new version of ORCHIDEE) |
| MH_L11Aer (S_Sr02) | As L11, but only dust and sea-salt considered in the aerosol forcing | Same as L11 |
| MH_L11AerEv (S_Sr03) | As L11aer, but with factor to limit bare soil evaporation | From year 250 of L11Aer |
| MH_FPMIP4 (S_Sr04) * | As L11AerEV, but with PMIP4 MH trace gazes and Earth's orbital parameters. Reference MH simulation without interactive vegetation | From year 250 of L11AerEv |
| Preindustrial (PI) | | |
| PI_PMIP3 | Reference PMIP3-CMIP5 IPSL simulation (Dufresne et al., 2013; Kageyama et al., 2013a) | |
| PI_FPMIP4 | As L11AerEV but with pre industrial trace gazes and Earth's orbital parameters | |


Table 1. Test done to set up the model with interactive vegetation. The different columns highlight the

name of the test and the initial state to better isolate the different factors contributing to the adjustment curves in
Figure 1. The simulations with an * are considered as reference for the model version and the transient
simulations. We include in parenthesis the tag of the simulation that corresponds to our internal nomenclature for
memory.




| Simulation | Comment | Initial state |
|---|---|---|
| **Mid Holocene (MH)** | | |
| MH-Vnone (V-Sr09) | L11Aer configuration but initial state with bare soil everywhere | Year 250 of L11Aer |
| MH-Vnone_FPMIP4 (V-Sr12)* | Same simulation as MH-Vnone, but using the PMIP4 trace gazes forcing | Year250 of MH-Vnone |
| MH-Vmap (V_Sr10) | As L11Aer, but vegetation map and soil initial state from an off line ORCHIDEE vegetation force with L11 pre-industrial simulation | Year 250 or L11Aer |
| MH-Vmap_FPMIP4 (V_Sr11) | Same simulation as MH-Vmap, but using the PMIP4 trace gazes forcing | Year 200 of MH-Vmap |
| **Pre Industrial (PI)** | | |
| PI-Vnone (V_Sr12) * | Preindustrial simulation corresponding to the MH simulations starting from bare soil | Year 500 of MH-Vnone-FPMIP4 |
| PI-Vmap (V_Sr07) | Preindustrial simulation corresponding to the MH simulation starting from the off line ORCHIDEE vegetation force with L11 pre-industrial simulation | Year 250 of Vmap_FPMIP4 |
| Transient 6000 BP – 0 BP | | |
| TRHOLV | Transient mid Holocene to present day simulation with dynamical vegetation | Year 500 of MH-Vnone-FPMIP4 |


Table 2. Simulations run to initialize the dynamical vegetation starting from bare soil or from a
vegetation map and soil moisture resulting from an off line ORCHIDEE simulation with dynamical vegetation
switch on and using the PI L11 simulated climate as boundary conditions. Simulations with an * are considered
as references for the model version and the transient simulations. We include in parentheses the tag of the
simulation that corresponds to our internal nomenclature for memory.



**7    Figure Caption**

Fig. 1: Mid Holocene annual mean precipitation (mmd⁻¹) and 2m air temperature (°C) differences

between a) FPMIP4 and PMIP3, b) L11 and PMIP3, c) L11Aer and L11 and d) L11AerAV and L11Aer. See
Table 1 and text for the details about the different simulations.

Figure 2: Illustration of the effect of the different adjustments made to produce mid-Holocene

simulations with the modified version of the IPSLCM5A-MR version of the IPSL model in which the land
surface model ORCHIDEE includes a different soil hydrology and snow models (see text for details). The three
panels show the global average of a) net surface heat flux (W.m⁻²), b) evaporation (kg.m⁻²), and c) 2m air
temperature (°C). The different color lines represent the results for the different simulations reported in Table 1.

Figure 3. a) Annual mean global model bias (bias_xy) and b) spatio-temporal root mean square

differences (rms_xyt) computed on the annual cycle (twelve climatological months) over the globe for the
different pre-industrial simulations considered in this manuscript (colors lines) and individual simulations of the
CMIP5 multi-model ensembles (grey lines). The metrics for the different variables are presented as parralel
coordinates, each of them having their own vectical axis with corresponding values. In these plots ta stands for
temperature (°C) with s for surface, 850 and 200 for 850 and 300 hPa, prw for total water content, pr for
precipitation (mmd⁻¹), rlut, for outgoing long wave radiation, rltcre and rltcre for the cloud radiative effect at the
top of the atmosphere in the short wave and long wave radiation respectively (Wm⁻²).

Figure 4. Long term adjustment of vegetation for mid Holocene when starting from bare soil (Vnone) or

from a vegetation map (Vmap). The 13 ORCHIDEE PFT have been gathered as bare soil, grass, tree and land-
use. When the dynamical vegetation is active only natural vegetation is considered. Land-use is thus only present
in one simulation, corresponding to a pre-industrial map used as reference in the IPSL model (Dufresne et al.
2013). The corresponding vegetation is referred to as PI_prescribed. Following Table 2, MH and PI refer to
midHolocene and Pre industrial control simulations respectively. The x axis is in months, starting from 0.

Figure 5: Vegetation maps obtained with the two different initial states for a) d) g) mid Holocene

simulations, b) e) h) pre-industrial simulations and c) f) i) pre-industrial simulation for Vnone. Vmap stands for
simulations where the mid-Holocene vegetation has been initialized from a vegetation map and Vnone for
simulations where the mid-Holocene has been initialized from bare soil.  For simplicity we only consider
fractions of a) b) c) bare soil, d) e) f) grass and g) h) i) trees.

Figure 6: Impact of the dynamical vegetation and initialization of vegetation on the simulated climate.

Differences for annual mean a) c) e) precipitation (mm.d⁻¹) and b) d) f) 2m air temperature (°C) between a) and b)
the mid Holocene simulation with dynamical vegetation (MH-VNone) and the mid Holocene simulation without
(MH FPMIP4), d) and d)  the mid Holocene (MH-Vnone)  and the pre-industrial (PI-Vnone) simulations with
bare soil as initial state for vegetation, and e) and f)  the two pre-industrial simulations initialized from bare soil
(PI-Vnone) or a vegetation map for vegetation (PI-Vmap). See table 2 and text for details on the simulations.





Figure 7: (a) Simulated megabiome distribution by MH_Vnone, converted from the modelled PFT
properties using the default algorithm described in Figure A1. (b) Reconstructions in BIOME 6000 DB version 1
(Harrison, 2017). (c) Number of pixels where reconstruction is available and the model matches (or does not
match) the data. Note that multiple reconstruction sites may be located in the same model grid cell, in which case
we did not group them so that each site was counted once. Numbers in parenthesis on the x axis in c) represent
the number of sites for each biome type.

Figure 8: Comparison of the change in vegetation between mid Holocene and preindustrial climate in
the two sets of experiments where the only difference is the way vegetation has been initialised for the mid-
Holocene simulation.  In a) c) e) Vmap correspond to simulatons where the MH simulation has been initialized
from a map and in b) d) f) Vnone to simulations where it has been initialized from baresoil. For simplicity we
only consider fractions of a) b) bare soil, c) d) grass and e) f) trees.

Figure 9: Evolution of trace gazes : $CO_2$ (ppm), $CH_4$ (ppb) and $N_2O$ (ppb), following Otto-Bliesner et al.
(2017).


Figure 10. Long term evolution of incoming solar radiation at the top of the atmosphere (TOA)($Wm^{-2}$,
top panel) and associated response of temperature (°C) and precipitation (mm.$y^{-1}$) expressed as a difference with
the 6000 year PB initial state and smoothed by a 100 year running mean) for a) NH Summer, b) Northern
hemisphere winter, c) Southern Hemisphere summer, and d) Southern Hemisphere winter. Temperatures are
plotted in red and precipitation in blue for summer, and they are respectively plotted in orange and green for
winter.  NH Summer and SH Winter correspond to June to September averages whereas NH winter and SH
summer correspond to December to March averages. All curves, except insolation have been smoothed by a 100
year running mean.

Figure 11: Long term evolution of the simulated a) baresoil, b) grass and c) tree covers, expressed as the
percentage (%) of Global, Norther Hemisphere or Southern Hemisphere continental areas, and d) GPP (PgC/y)
over the same regions.  Annual mean values are smoothed by a 100 year running mean.

Figure 12 : Long term evolution of Baresoil, grass and Tre, expressed as the % of land cover North of
60°N,  over Eurasia and over West Africa. The different values are plotted as differences with the first 100 year
averages.  A 100 year running mean is applied to the curves before plotting.

Figure 13. Long term evolution of total GPP (PgC/y for land surfaces north of 60°N (blue) Eurasia
(cyan), and W Africa (pink). Annual mean values are smoothed by a 100 yr running mean.

Figure 14: Vegetation map comparing a) the Mid Holocene  (1[st] 50 years) and the pre-industrial (50
year around 1850 AC (last 150 to 100 years)  periods of the transient simulation , b) the difference between pre-
industrial climate for the transient simulation and the Vnone simulations, and c) the differences between the





historical period (last 50 years) and the pre-industrial period of the transient simulation. For simplicity we only
consider bare soil (top), grass (middle) and tree (bottom).

Figure 15 : Evolution of the different tree PFTs in Eurasia, expressed as the percentage change

compared to their 6000 year BP initial state.. Each color line stands for a different PFT. Values have been
smoothed by a 100 year running mean.








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



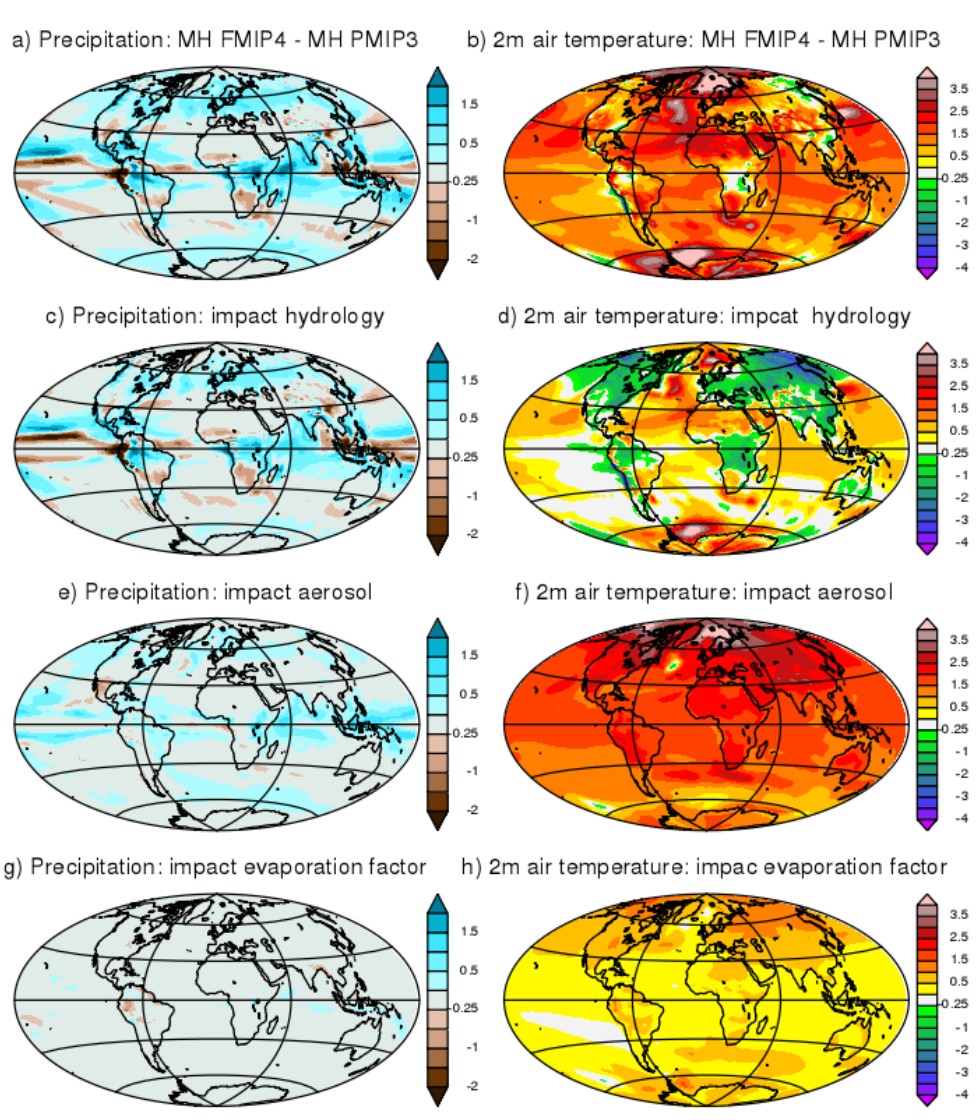

Fig. 1: Mid Holocene annual mean precipitation (mmd$^{-1}$) and 2m air temperature (°C) differences between a)
FPMIP4 and PMIP3, b) L11 and PMIP3, c) L11Aer and L11 and d) L11AerAV and L11Aer. See Table 1 and text for
the details about the different simulations.





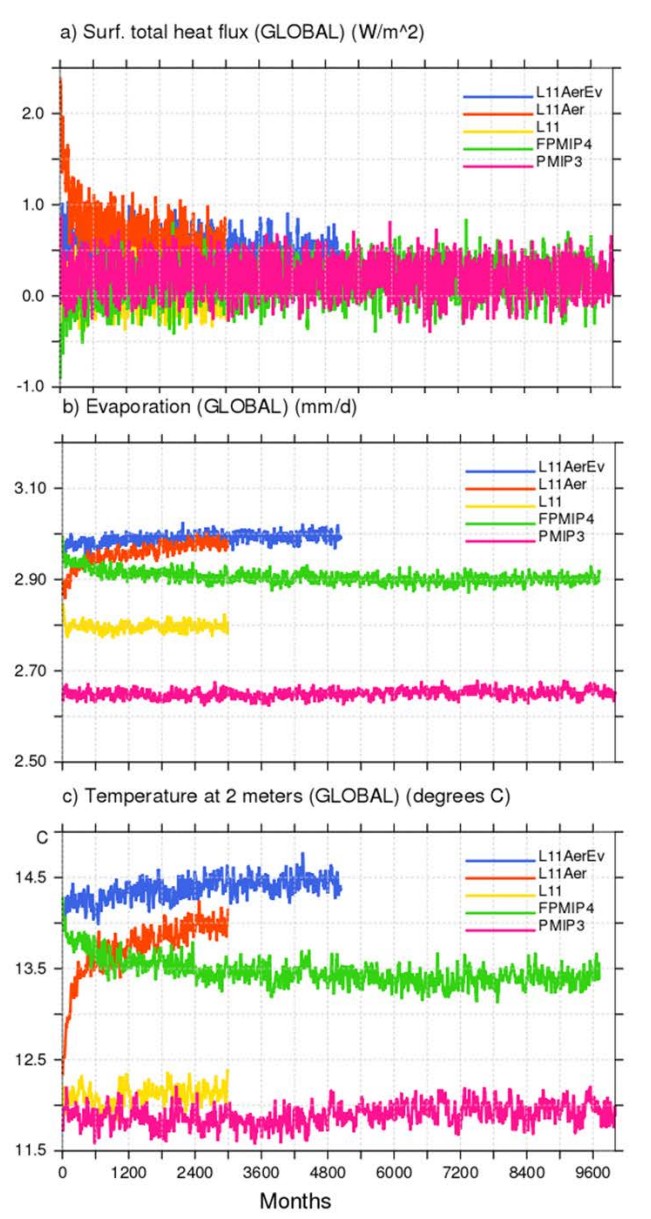

Figure 2: Illustration of the effect of the different adjustments made to produce mid-Holocene simulations with
the modified version of the IPSLCM5A-MR version of the IPSL model in which the land surface model ORCHIDEE
includes a different soil hydrology and snow models (see text for details). The three panels show the global
average of a) net surface heat flux (W.m$^{-2}$), b) evaporation (kg.m$^{-2}$), and c) 2m air temperature (°C). The
different color lines represent the results for the different simulations reported in Table 1.

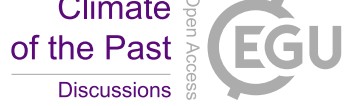









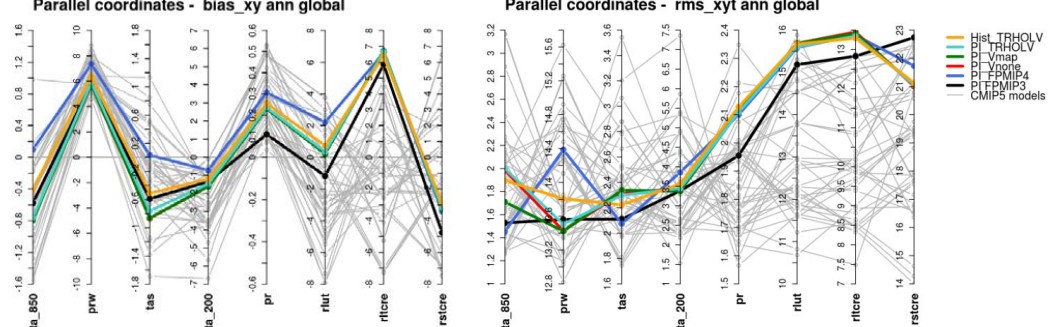


Figure 3. a) Annual mean global model bias (bias_xy) and b) spatio-temporal root mean square differences (rms_xyt) computed on the annual cycle (twelve climatological months) over the globe for the different pre-industrial simulations considered in this manuscript (colors lines) and individual simulations of the CMIP5 multi-model ensembles (grey lines). The metrics for the different variables are presented as parralel coordinates, each of them having their own vectical axis with corresponding values. In these plots ta stands for temperature (°C) with s for surface, 850 and 200 for 850 and 300 hPa, prw for total water content, pr for precipitation (mmd$^{-1}$), rlut, for outgoing long wave radiation, rltcre and rltcre for the cloud radiative effect at the top of the atmosphere in the short wave and long wave radiation respectively (Wm$^{-2}$).






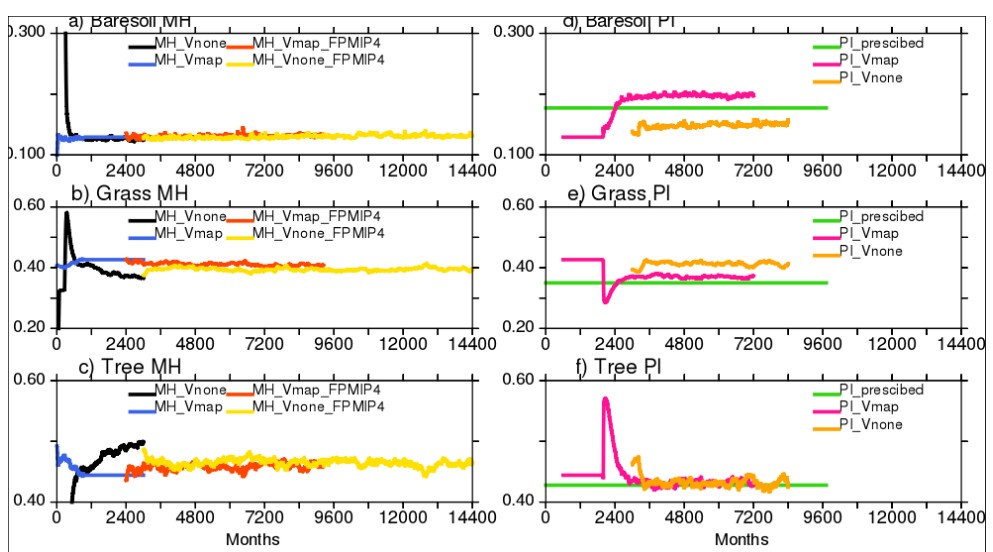

Figure 4. Long term adjustment of vegetation for mid Holocene when starting from bare soil (Vnone) or from a
vegetation map (Vmap). The 13 ORCHIDEE PFT have been gathered as bare soil, grass, tree and land-use. When
the dynamical vegetation is active only natural vegetation is considered. Land-use is thus only present in one
simulation, corresponding to a pre-industrial map used as reference in the IPSL model (Dufresne et al. 2013).
The corresponding vegetation is referred to as PI_prescribed. Following Table 2, MH and PI refer to
midHolocene and Pre industrial control simulations respectively. The x axis is in months, starting from 0.

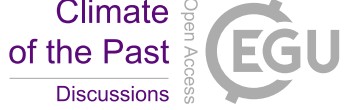




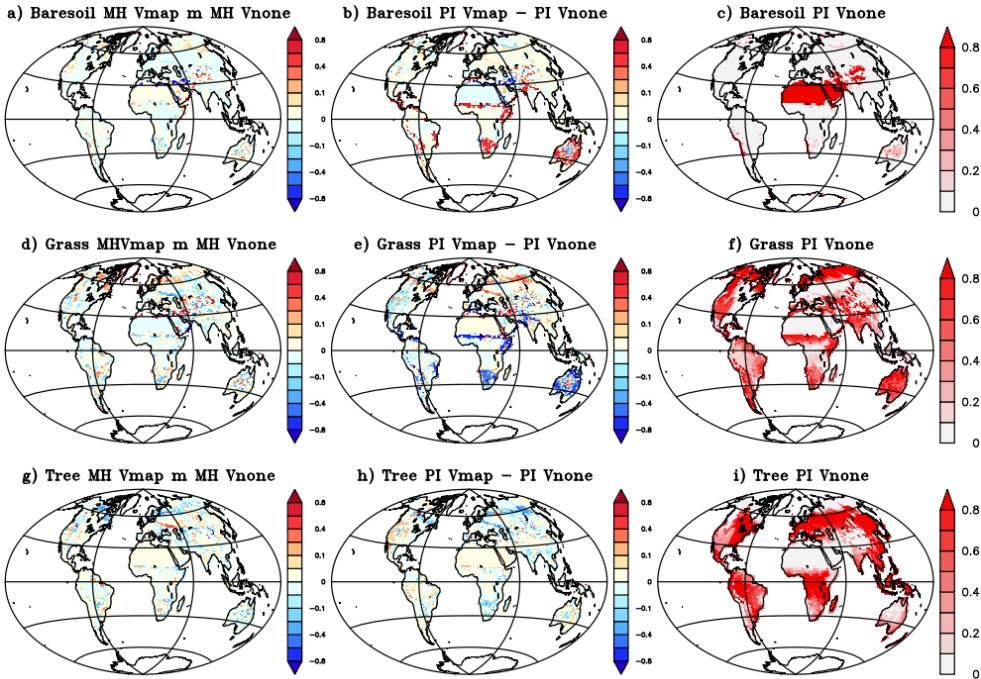

Figure 5: Vegetation maps obtained with the two different initial states for a) d) g) mid Holocene simulations,
b) e) h) pre-industrial simulations and c) f) i) pre-industrial simulation for Vnone. Vmap stands for simulations
where the mid-Holocene vegetation has been initialized from a vegetation map and Vnone for simulations
where the mid-Holocene has been initialized from bare soil. For simplicity we only consider fractions of a) b) c)
bare soil, d) e) f) grass and g) h) i) trees.





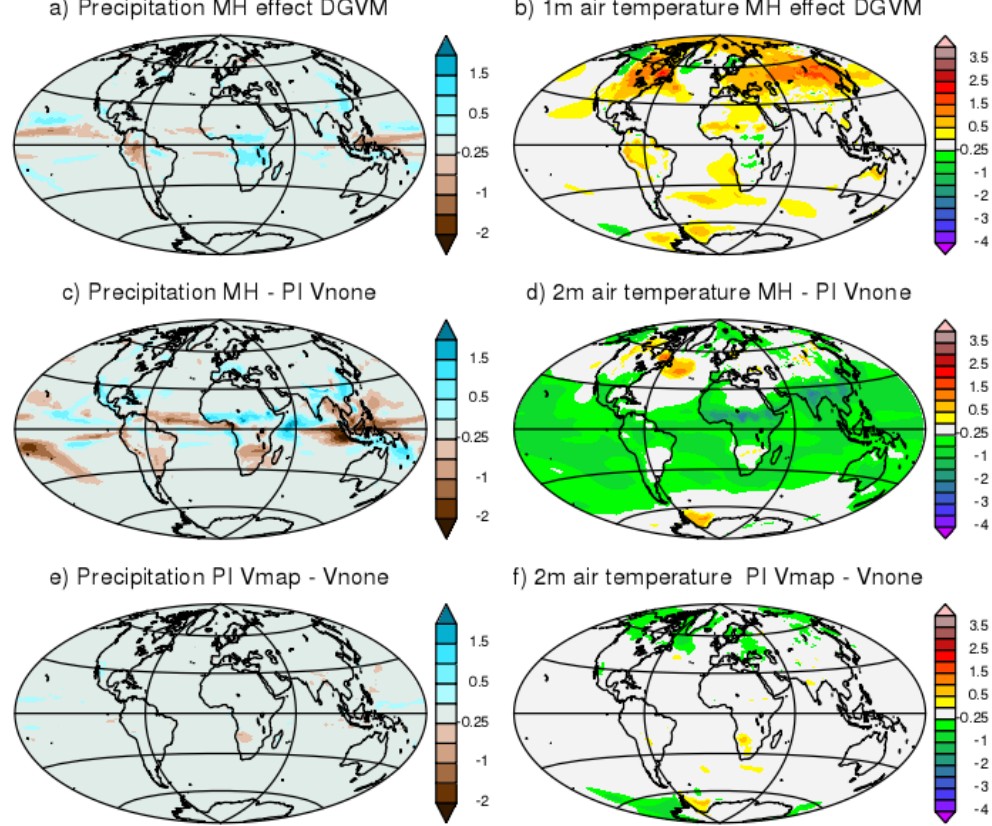

Figure 6: Impact of the dynamical vegetation and initialization of vegetation on the simulated climate.
Differences for annual mean a) c) e) precipitation (mm.d-1) and b) d) f) 2m air temperature (°C) between a) and
b) the mid Holocene simulation with dynamical vegetation (MH-VNone) and the mid Holocene simulation
without (MH FPMIP4), d) and d)  the mid Holocene (MH-Vnone)  and the pre-industrial (PI-Vnone) simulations
with bare soil as initial state for vegetation, and e) and f)  the two pre-industrial simulations initialized from
bare soil (PI-Vnone) or a vegetation map for vegetation (PI-Vmap). See table 2 and text for details on the
simulations.




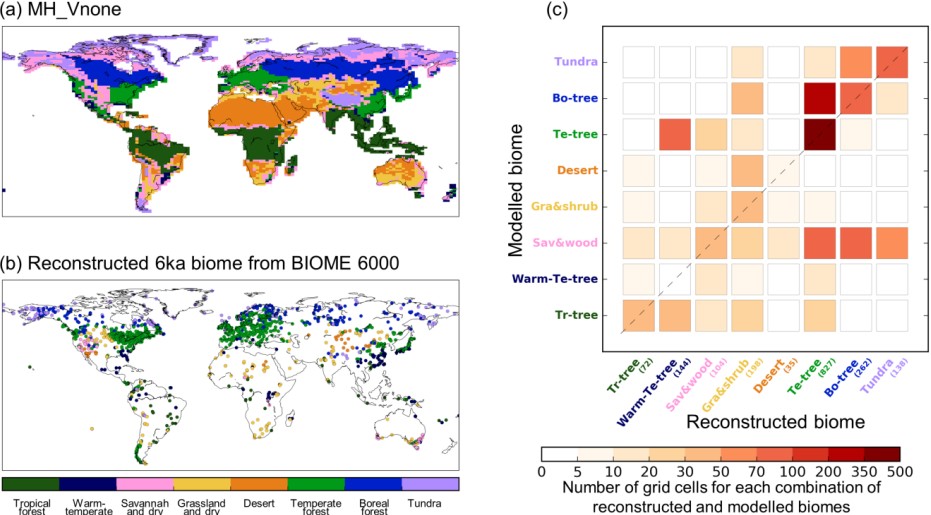

Figure 7: (a) Simulated megabiome distribution by MH_Vnone, converted from the modelled PFT properties
using the default algorithm described in Figure A1. (b) Reconstructions in BIOME 6000 DB version 1 (Harrison,
2017). (c) Number of pixels where reconstruction is available and the model matches (or does not match) the
data. Note that multiple reconstruction sites may be located in the same model grid cell, in which case we did
not group them so that each site was counted once. Numbers in parenthesis on the x axis in c) represent the
number of sites for each biome type.





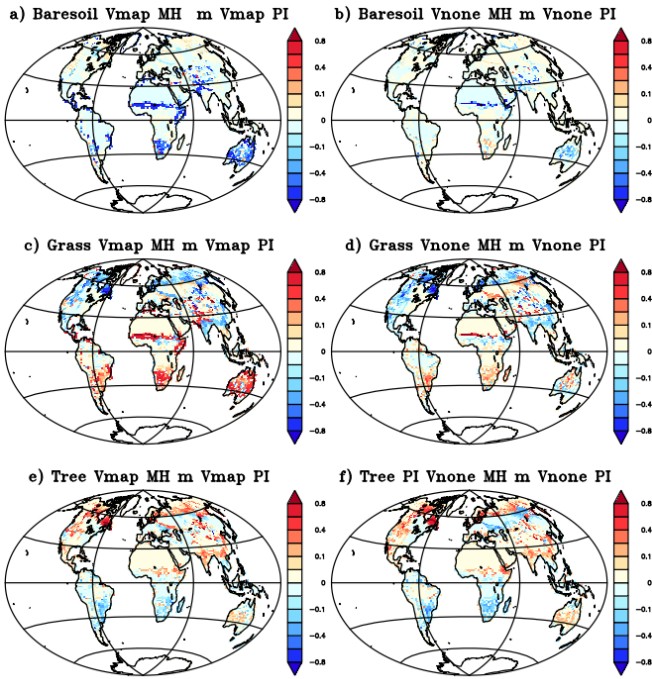

Figure 8: Comparison of the change in vegetation between mid Holocene and preindustrial climate in the two
sets of experiments where the only difference is the way vegetation has been initialised for the mid-Holocene
simulation. In a) c) e) Vmap correspond to simulatons where the MH simulation has been initialized from a
map and in b) d) f) Vnone to simulations where it has been initialized from baresoil. For simplicity we only
consider fractions of a) b) bare soil, c) d) grass and e) f) trees.







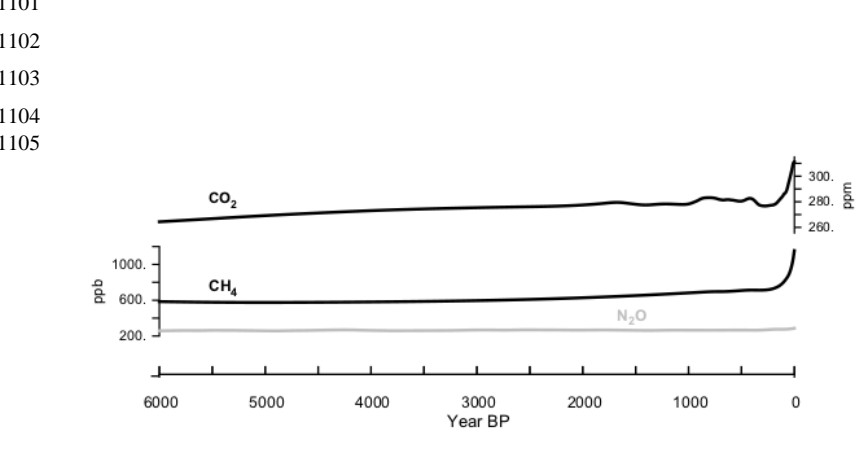

Figure 9: Evolution of trace gazes : $CO_2$ (ppm), $CH_4$ (ppb) and $N_2O$ (ppb), following Otto-Bliesner et al. (2017).





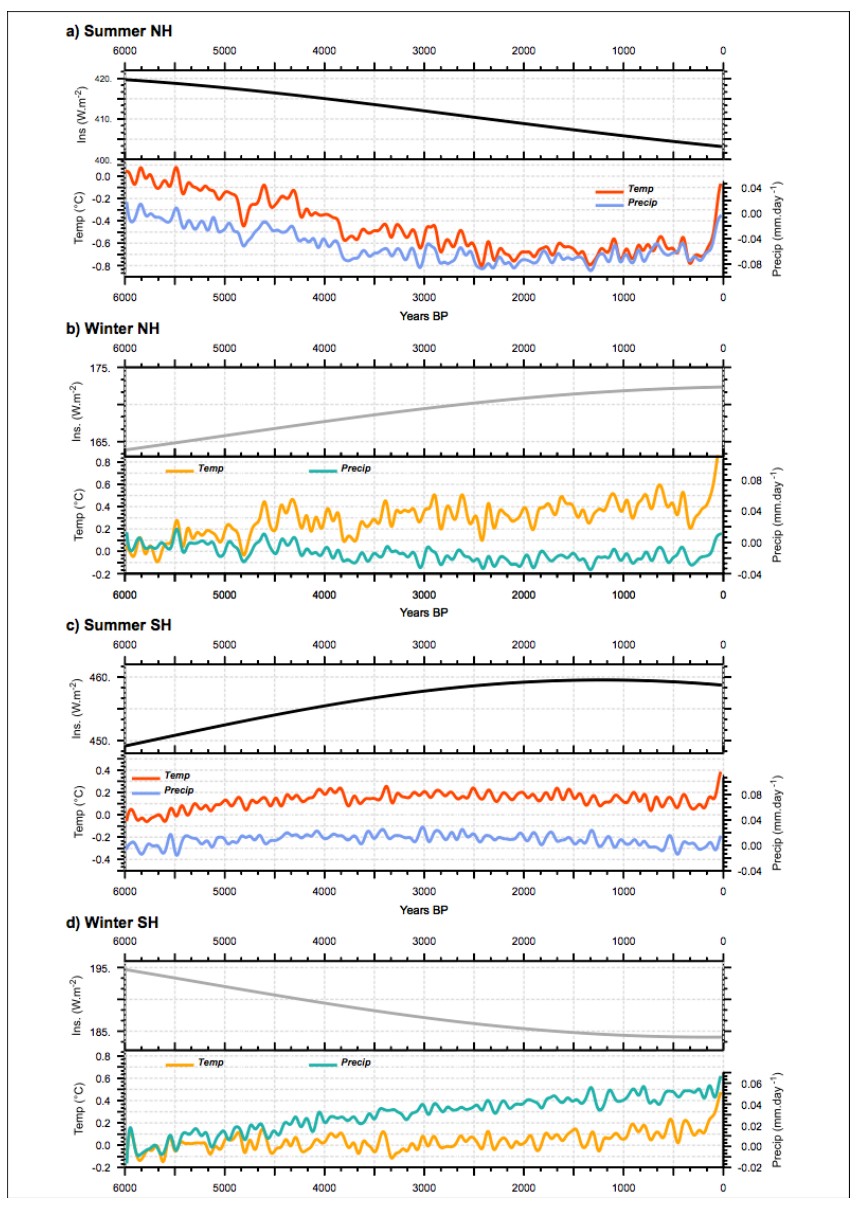


Figure 10. Long term evolution of incoming solar radiation at the top of the atmosphere (TOA)(Wm$^{-2}$, top panel)
and associated response of temperature (°C) and precipitation (mm.y$^{-1}$) expressed as a difference with the
6000 year PB initial state and smoothed by a 100 year running mean) for a) NH Summer, b) Northern
hemisphere winter, c) Southern Hemisphere summer, and d) Southern Hemisphere winter. Temperatures are
plotted in red and precipitation in blue for summer, and they are respectively plotted in orange and green for
winter.  NH Summer and SH Winter correspond to June to September averages whereas NH winter and SH
summer correspond to December to March averages. All curves, except insolation have been smoothed by a
100 year running mean.





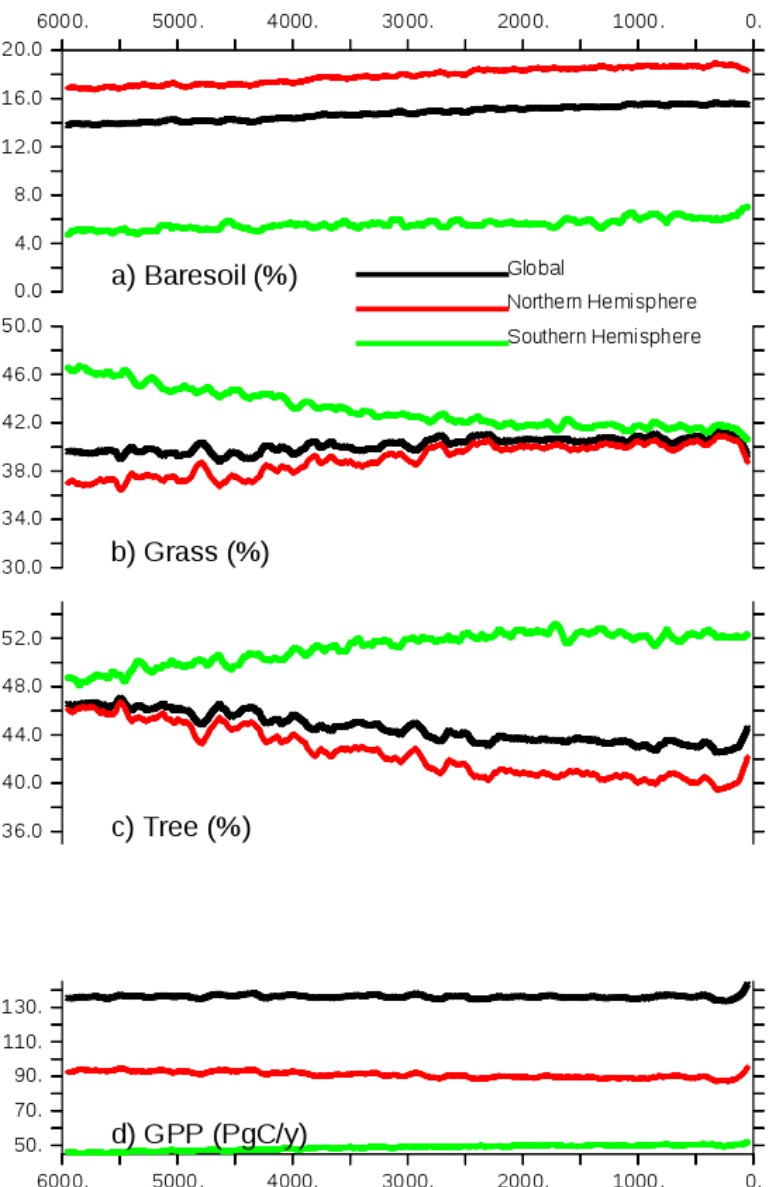


Figure 11: Long term evolution of the simulated a) baresoil, b) grass and c) tree covers, expressed as the
percentage (%) of Global, Norther Hemisphere or Southern Hemisphere continental areas, and d) GPP (PgC/y)
over the same regions.  Annual mean values are smoothed by a 100 year running mean.

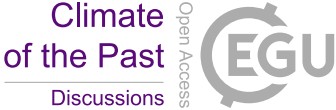

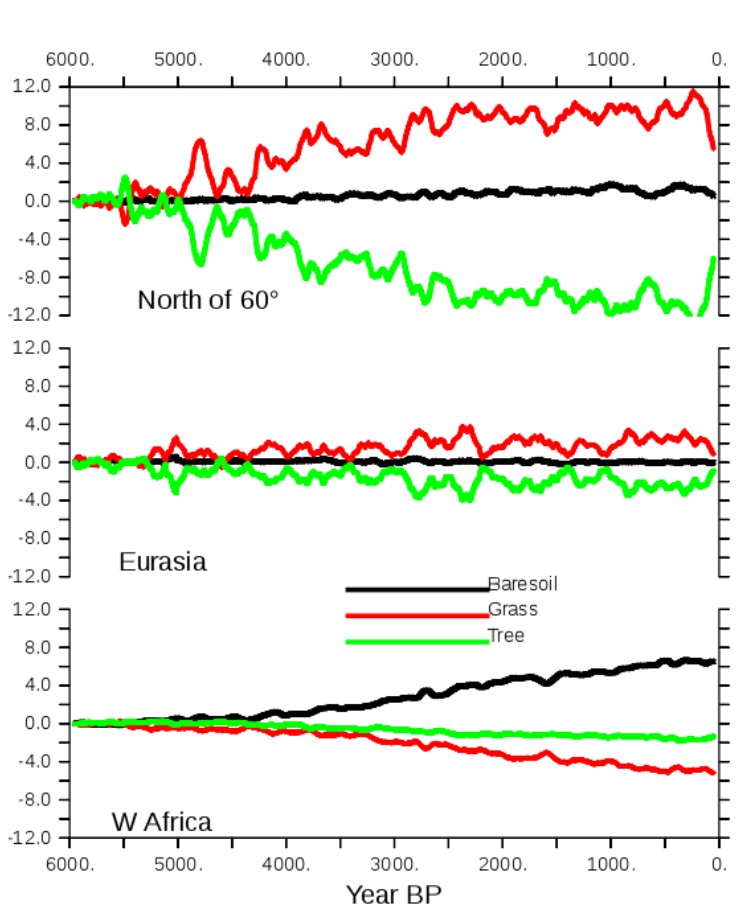


Figure 12 : Long term evolution of Baresoil, grass and Tre, expressed as the % of land cover North of 60°N,  over
Eurasia and over West Africa. The different values are plotted as differences with the first 100 year averages.  A
100 year running mean is applied to the curves before plotting.






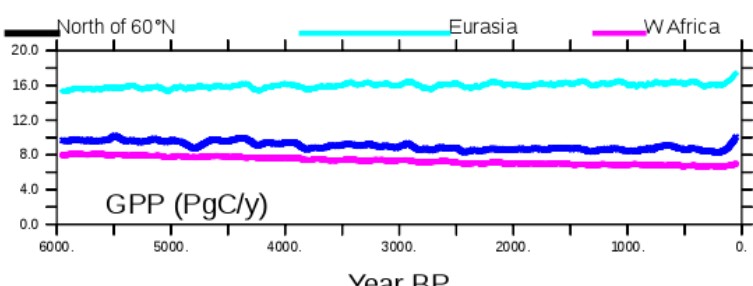

Figure 13. Long term evolution of total GPP (PgC/y for land surfaces north of 60°N (blue) Eurasia (cyan), and W
Africa (pink). Annual mean values are smoothed by a 100 yr running mean.








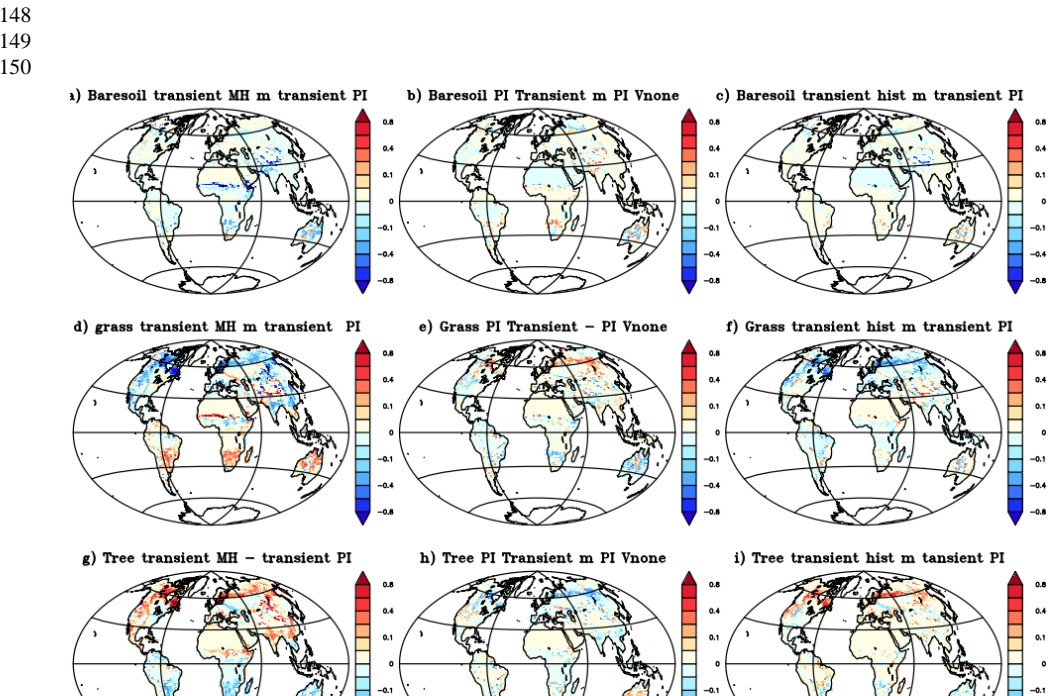

Figure 14: Vegetation map comparing a) the Mid Holocene (1st 50 years) and the pre-industrial (50 year around
1850 AC (last 150 to 100 years) periods of the transient simulation , b) the difference between pre-industrial
climate for the transient simulation and the Vnone simulations, and c) the differences between the historical
period (last 50 years) and the pre-industrial period of the transient simulation. For simplicity we only consider
bare soil (top), grass (middle) and tree (bottom).





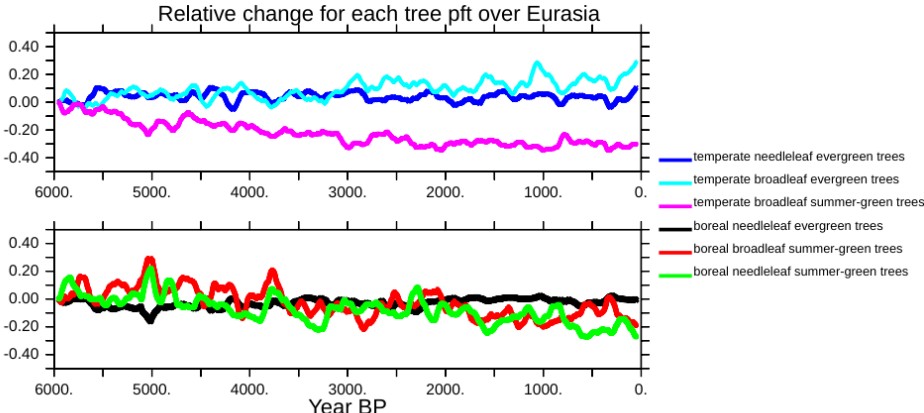


Figure 15 : Evolution of the different tree PFTs in Eurasia, expressed as the percentage change compared to
their 6000 year BP initial state.. Each color line stands for a different PFT. Values have been smoothed by a 100
year running mean.