# Peer review of "Strength and challenges for transient mid to late Holocene simulations with dynamical vegetation"

_Climate of the Past, 2018_

## Author Comment (AC1) · 30 Nov 2018

Dear editor,

You raised the point that figure A1 and A2 were missing in the manuscript we submitted. On behalf of the co-authors, I would like to apology for it. I made an error when merging the files to create the pdf file we submitted. I therefore attach a complete manuscript with all figures to this comment, so that all elements can be considered in the review process.

Best regards Pascale Braconnot

Please also note the supplement to this comment:

[Figure]

https://www.clim-past-discuss.net/cp-2018-140/cp-2018-140-AC1-supplement.pdf

[Figure]

**Supplement:**

[revised manuscript text omitted]

---

## Referee Comment (RC1) · Anonymous Referee #1 · 4 Dec 2018

In their manuscript, the authors not only present the first transient Holocene simulation with the IPSL Earth System Model, but also a set of different time-slice experiments that has been used to initialize and evaluate the transient simulation and to test the stability of the simulated vegetation distributions. The transient simulation captures the main Holocene long term vegetation trends reasonably well and show regionally different timings of major vegetation changes. It indicates large scale vegetation degradation. The integration of several different tree PFT types enables the model to also reveal strong changes in the Eurasian forest composition. The sensitivity experiments show that the calculated vegetation, temperature and precipitation distribution depend on the chosen model setup. Initializing the model with totally different vegetation states (bare soil everywhere vs. simulated vegetation map) leads only to slight differences

for mid-Holocene boundary conditions, but to multiple states under pre-industrial conditions. The author raise the question, whether the vegetation-climate system is more sensitive in pre-industrial times. The large number of experiments around the actual transient simulation indicates that the authors have designed this study very carefully and thoughtfully. They raise a couple of interesting questions and show what should be considered when setting up and evaluating transient simulations. The single results are presented clearly and accurately and are accompanied by many well-designed figures. The study is interesting and well suited to be published in Climate of the Past, though I suggest few fundamental revisions.

My main point is that the main objectives of the study remain unclear. The authors analyze the effects of a variety of changes in the model and setup of the simulations compared to the CMIP5 model-version, but it is not necessarily clear why these changes are chosen and why they are relevant for the transient simulation. Furthermore, the results (also those considering the transient simulation) are usually only mentioned because a detailed analysis of the effects would go beyond the scope of a single study and requires even more simulations. This gives you the feeling that everything is addressed, but without a clear outline and without a clear (final) message. There are several points that reinforce this impression:

- The single chapters are not really linked and there is no separate method chapter, instead methods and results are merged in the individual sections.

- you somehow get lost in the different simulations and don't always know which simulation is being talked about or what are the characteristics of this simulations (unless you look it up again).

- The title is 'strength and limits of transient...Holocene simulations' and also the abstract promises a focus on the transient simulation, but then, 9 pages follow discussing other simulations. At page 10; the Holocene simulation is introduced followed by only 3 pages presenting the results of this transient simulation. I think this ratio of results

from the transient simulation to the results from the other simulations is not balanced, particularly because a clear connection to the results of the other simulations may not always be seen.

- I furthermore think that the conclusions do not represent the title. What are the strength and limits of the transient simulation? From reading the conclusions I do not know this...

I recommend to restructure the manuscript. The time-slice experiments must be embedded more strongly in the results of the transient simulation and a clear link must be established between the simulations. To reduce the number of experiments and figures, the simulations dealing with finding an appropriate initial state or discussing the differences to the PMIP3-CMIP5 model could be shifted to the Appendix. These are technically interesting but seem not to follow any scientific question. The result section of the transient simulations should be extended and more specified. In addition, research questions and aims of the study should be worked out to give the results a clear framework.

Minor revisions and technical corrections:

- The abstract should focus more on the highlights of the results.

- L36: write: reduction in North African monsoon intensity. . .

- L38: separate 'variabilityor' in 'variability or'

- L56: Do you mean 'mid-Holocene boundary conditions' ?

- L94-95: The sentence 'before. . .' needs a verb.

- L111: It is '30min'

- L130: what do you mean with 'transient late Holocene simulation'

- L145: it is 'trace gases', please correct it in the entire manuscript

- L160: Again, what is meant by 'transient late Holocene simulation'

- L178: delete the first 'close'

- L185: Do you mean 'land surface cooling is due to? . . . ('is' instead of 'as')

- L185: 'first L11 version' why first?

- L202: simulations of global warming

- L202: There is no 'Fig. 2e'

- L203-205: I did not get these sentences, please specify.

- L243: Please explain the metrics in more detail (e.g. in the Appendix) because the metric package may be unknown to the readers.

- L257-258: To me it is not clear which simulations ran with dynamic vegetation and which not.

- L263: it is '1901-1910'

- L314: The heading of chapter 3 is: 'mid-Holocene simulations. . .' so why is there a section dealing with pre-industrial climate?

-L319-320: I do not understand what is meant by 'vegetation biases' in this context. When vegetation is interactive, the calculated vegetation distribution can be biased, but how does this bias impact the representation of the simulated vegetation? Please clarify.

-L337: What do you mean by this? that the differences in PI simulations are of similar magnitude as the differences between PI and MH ?

-L347: delete the blank after NH.

-L353: 'no' instead of 'not'

-L383: '. . . follow the long term insolation changes in each hemisphere. . .' What about

SH Winter? Please be more precise.

-L:385: Do you mean 'MH insolation forcing' ?

-L389: this 'plateau' trend in temperature during SH summer is not a feature of the insolation forcing, showing still an increase in insolation during this period. For the NH winter temperature change, variability is very strong, but the main trend is still increasing and not 'plateau' shaped.

-L409: It seems as if the tree fraction follows the summer insolation change. Please specify and explain. What about the annual mean changes in temperature, precipitation and insolation?

-L435-436: Where does this variability come from?

-L437-438: Something is wrong in this sentence, I guess the first 'and'.

-L456: What do you mean by 'rapid changes' and if these 'deserve attention' why don't you investigate them in this study?

-L459: 'several' instead of 'sever'

-L462: 'Is it the case. . .' instead of 'It is the case. . .'

-L476-477: Is there a possibility to figure out the reasons for having different PI climate-vegetation changes?

-L493: 'including instead of 'to include'

-L496: I suggest adding: 'large scale features in climate and vegetation change expected from . . .'

-L499: 'annual mean' of what?

-L502: 'Why should the trend be linear?

-L514: 'It might. . .' instead of 'I might'

[Figure]

-L528-530: This needs to be further specified.

-L535: Maybe it is 'other' instead of 'over'

-L544: Is it really 'reference period' or just 'reference simulation' ?

-L581: Why is 'BIOMES' written in capital letters?

-L581: Isn't it originally the method of Prentice et al.2011? What is different to the method of Zhu et al. 2018?

-L589: it is: high northern latitudes

-L590: It is not obvious why the GDD limit of 500°C is being tested, are these values realistic? I guess ORCHIDEE also uses a GDD limit of > 350°C for the existence of boreal trees vs. tundra (GDD5<350°C). A biomisation using a GDD limit of 500°C thus may not represent the vegetation simulated by the model, because it suggest tundra in regions that are suited for forests.

-L595: Something is wrong in this sentence. . .

-L598-599: Should we now reconsider the choice of bioclimatic limits in the DGVMs?

- What about data availability?

-Fig 2: The units in the caption do not agree to the units in the headings of the figures

-Fig.3: The pressure levels do not match, please also explain the metrics in more detail.

-Fig.4: Why are MH_Vnone and MH_Vmap so different?

-Fig.5: In the headings for a) d) and g) you use m instead of minus, please change

-Fig.7: It should be explained, why there is Savanna in the northern latitudes. In my print, the pink and orange color is not really distinguishable. Please state, why there is no grassland in North Africa in your simulation

-Fig.9: Maybe this figure could be moved to the Appendix.

-Fig.10: I think it is very interesting, that the winter precipitation on the NH does not (or only very slightly) changed during the Holocene.

-Fig.10: L1116: NH summer, b) Northern hemisphere winter . . .. you could also write NH winter, furthermore, winter and summer season should be defined in the caption (i.e. months that are taken)

-Fig.10: When looking into palaeo-seasons, one always faces the problem of different calendars. The months NDJF or JJA differ in length between mid-Holocene and PI. It should at least be mentioned in the text and in the caption, that this 'problem' exist and is not considered, neither by the model nor in the analysis. But this problem may change the trends discussed here!

-Fig.11: it is 'Northern Hemisphere'. It would also be interesting, how the simulated tree cover and bare-soil fractions at the end of the simulation compare to modern estimates on (natural) tree cover. How large is the underestimation of forest in the high northern latitudes by the model?

Fig.12: What causes the strong peak around 4.8ka ?

Fig.12: L1128: 'tree' instead of 'Tre'

Fig.14: 'm' vs 'minus' or '-', and sometimes bare soil is written in two words sometimes not.

-L1164 delete one '.'

Fig.A2: citation for the BIOME6000 records is needed

---

## Referee Comment (RC2) · Anonymous Referee #2 · 6 Dec 2018

Summary:

Braconnot et al. use the IPSL Earth system model with dynamic vegetation to perform a transient simulation from the mid-Holocene to present-day. The authors begin by testing how their changes to the default IPSL model impact the mean climate state. Modifications are made to the land surface model, aerosol emissions, soil evaporation, and vegetation. Sensitivity tests show some significant changes in the mean temperature and precipitation associated with their model modifications. Surprisingly, overall biases do not change much from the default IPSL model configuration. The authors then perform different vegetation initialization experiments and discover a single equilibrium state (more or less) for the mid-Holocene but multiple equilibrium states for the preindustrial. Finally, they perform a transient simulation from the mid-Holocene to

present-day and compare model output with vegetation records.

Although there was clearly a lot of effort put into these simulations, many of the results are muddled by poor structuring and limited analysis. In its current form, the paper is not especially satisfying from a model development perspective or a paleoclimate perspective. Given the choice of journal, I recommend modifying the structure and results to better reflect the target audience. You can leave the model development and testing sections, but you need to provide additional context and analyses.

General Comments:

The results and analyses left me unsatisfied, especially given the amount of time spent on model testing. Many findings are dismissed as beyond the scope of this paper or for future work. However, without in-depth exploration of at least some of the interesting results of the simulations, the paper feels more like a data description, which is fine, but not especially appropriate for Climate of the Past. I recommend expanding the transient simulation results and analysis, since it is the novel part of this study. There are several topics that could be explored further, such as the importance of dynamic vegetation in the transient climate response (needed for the title), the mechanisms driving multiple equilibria, and comparison with proxy reconstructions. The authors might also want to consider how do these results compare with other transient model simulation? I have not edited this manuscript for spelling and grammar. There are many instances of incorrect spelling. Please proof-read the text and figure labels before resubmission. Also, I encourage the authors to seek assistance from a very proficient or native English speaker.

Specific Comments:

Line 123: Can sea level actually change in the model?

Line 143: I do not think that this is very good justification for not thoroughly testing modifications against preindustrial climate.

[Figure]

Lines 152-153: Given the importance of the aerosol responses, why do you prescribe aerosols here? Are dust and sea-salt prescribed to PI? How might this impact climate? I do not find ". . .we also plan to run simulations with fully interactive dust and sea-salt" good justification.

Line 163: Some of these modifications do not feel robust (e.g. the soil evaporation factor).

Lines 175-176: What is the TOA energy imbalance for these runs? This could be important since different simulations are run for different amounts of time. 0.4 Wm-2 is far from zero. . .

Line 181: Why does this modification impact the ocean response so dramatically? I thought the hydrologic modification should only impact the land surface. Am I missing something?

Lines 203-205: This sentence is ambiguous. What does ". . .might be small. . ." mean?

Line 213: "et al." for a personal communication?

Line 259: Is the carbon really in equilibrium?

Line 310: how well does 50% compare with other models?

Lines 325-326: "Since surface variables adjust rapidly, this is a way to compare the rapid adjustment to insolation

326 and the additional effect due to the dynamical vegetation (not discussed here)." Why say this then? It feels like an advertisement. . .

Line 367: Why not cite Joos?

Line 380: Do you mean JJAS? How do you account for the calendar changes?

Line 398: Interesting. . .Worth performing spectral analysis on the variability?

Lines 418-419: Why would this lead to an underestimate?

Line 445: "...provides the feeling..." is not scientific language.

Line 462-463: Is this sentence necessary?

Line 476-477: Could the difference relate to spin-up procedure?

Line 480: This kind of "additional analysis" is what is necessary to make this paper valuable to the community.

Figures:

For the difference maps, please center the contour ranges on 0 and add white in the middle to more easily distinguish differences. Change months on the X-axes to years. Figure 1: Add the PI runs to the comparison. Figure 6: Is it 1 m or 2 m air temperature? Figure 9: Needed? Could be supplemental.

---

## Author Comment (AC4) · 26 Jan 2019

Dear editor,

We have now posted the reply to the two comments of the reviewers. They include a proposition of what we can do to improve the story line of our manuscript and complement some of the analyses. We hope that you'll find these arguments convincing. With our best regards Pascale Braconnot on behalf of the co-authors.

---

## Author Response (AR1)

Dear Editor,

We have the pleasure to submit the revised version of the manuscript. We have implemented all the changes proposed in the response to the reviewers. The major changes concern a complete revision of the outline of the manuscript, while keeping its initial objectives. We also revisited the organization of the figures accordingly. We added figure 7 to enlarge the discussion on the linkages between the long term climate changes and the seasonal insolation forcing, figure 8 to show the changes in sea-ice and snow cover in the northern hemisphere, and figure 12 to quantify the southward shift of the African rainbelt and long term decrease in precipitation. We reduced the number of vegetation maps and keep those now in figure 10 to show the MH, PI and historical vegetation in the transient experiment, and in figure 16 to discuss the different vegetation states between our Vnone and Vmap simulations. A panel has been added in figure 5 to show the changes in insolation seasonality in the northern and the southern hemispheres. Figure 14 on the mega biome comparison now also include a comparison of the MH minus PI biomes.

We would like also to thank you for your comments. We provided a response to them below. We also include at the end of it the responses to the reviewers that are similar to the one we posted previously on the CP server.

We hope that you'll find the revision in agreement with the responses and in good shape for publication in climate of the past.

With our best regards

Pascale Braconnot, on behalf of the co-authors.

**Response to the editor comments.**
There are only a few items that I would like to add.

*One of your topics addresses multiple vegetation states. This is, indeed, a fascinating subject. You mentioned papers by Victor Brovkin and me. Hence it would be interesting to see whether you also find the differences in tropical large-scale circulation (e.g., a shift in the velocity potential) between the different states.*
We do not find big climate differences in climate. A robust feature is the small interhemispheric different in temperature and thus on cross equatorial heat transport. But it is tiny and not significant. The metrics also tell that it is not possible to statistically distinguish the two simulations, even when considering 2 regions and different seasons. Because of this we didn't add much on the analysis of the climate part. Future work would require running an ensemble to fully test this vegetation instability.

*In your reply to reviewer 1 you mentioned that you would use the biomisation method to evaluate the different vegetation states. Perhaps I just mis-understood this point. It would be more appropriate to compare simulated vegetation patterns directly instead of diagnosing these patterns by using the indirect biomisation.*

The biomisation method is only used to compare with BIOME6000 reconstruction. All maps showing baresoil, grass and tree are from the original ORCHIDEE PFTs. We also added a table in the appendix to address reviewer 1 comment, that we could have compared the simulated PI PFTs with those of the 1860 map we use when vegetation is prescribed considering grid points without land use. We also included a similar comparison for the grid points with land use, to show the differences in MH vegetation with the dynamical vegetation compared to the 1860 map. We also insist on the fact that the differences include both model biases and MH climate-vegetation feedback.

*Reviewer 2 questions the use of constant aerosol forcing. The effect of changes in Holocene aerosol concentration is a topical question. (You mentioned the papers by Francesco Pausata, for example.) It seems that it is not at all clear what effect a change in mineral dust during the Holocene has had on Holocene climate and climate variability. The effect of mineral dust on precipitation, for example, strongly depends on the chosen values of optical properties of the aerosols.*

The aerosols forcing we use is what is usually done in past climate simulations. There is nothing particular, except that we have to explain our choice to only consider dusts and sea-salt. We added a sentence to fully justify our choice by the fact that the effect we introduce by doing this on the mean climate is larger than the effect we expect from the change in MH dust. We agree that the question of MH dust is a complex topic. We mention several papers that consider dusts just to tell this is an open question. Pausata et al s study should be considered as an extreme oversimplified test case. It is interesting, but cannot be considered as realistic, but we made no particular comment in the text, because we are not looking at dust in details

*Reviewer 2 also mentions the calendar problem. Looking at the recent paper by Bartlein and Shafer (see GMD Discuss.), the effect can be pretty large, if monthly means are concerned. Averaging over 4 months should have a smaller effect. Perhaps you can cite your 1997 paper with Sylvie Joussaume?*

Thank you for this remark. Yes we added the reference to Joussaume and Braconnot 1997, and we hope it is clear that we only look at effects that would emerge whatever the calendar we use for this period.

*Line 35 of your manuscript: the term 'intertropical convergence zone' might be misleading, if you want to refer to the tropical rainbelt – see Sharon Nicholson's critique which appeared in BAMS, Feb. 2018, pp. 337.*
Normally now rain belt is used when referring to precipitation over land.

*Lines 69 ff: model biases: You are right that an atmosphere-ocean GCM is a different model than an atmosphere-ocean-vegetation model. Looking at the Holocene West African monsoon, I would argue that the biases of an atmosphere-ocean GCM are larger than that of an atmosphere-ocean-vegetation model. The former generally produces too small monsoon rain than the later. But can you really compare the biases of different model types?*
A difficult question. We are not able and do not do it. We only show how big they are using the metrics. This is an important point for model-data comparison, knowing that adding degrees of freedom in general degrade the simulated climatology, but not necessarily the mechanisms of

climate change. The last sentence of the text is on the need to develop methodologies to evaluate processes rather than climatological variables.

*Line 78: A word on why you start at 6ka would be sensible. It is the old problem: the Holocene climate around 6ka reveals pretty strong changes.*

We didn't comment much on this in the text, because we start from the well-established PMIP simulations. At least the ocean component is closer to present day conditions, so that we do not have to care too much of the ocean initial state, and 1000 year Mid-Holocene simulations were long enough to initialize the whole system.  We have this question somehow for any period outside the modern range.

**Response to Reviewer 1 comments**

Reviewer 1 provided several important comments on the structure and the objectives of the manuscript. The major recommendation is

*"I recommend to restructure the manuscript. The time-slice experiments must be embedded more strongly in the results of the transient simulation and a clear link must be established between the simulations. To reduce the number of experiments and figures, the simulations dealing with finding an appropriate initial state or discussing the differences to the PMIP3-CMIP5 model could be shifted to the Appendix. These are technically interesting but seem not to follow any scientific question. The result section of the transient simulations should be extended and more specified. In addition, research questions and aims of the study should be worked out to give the results a clear framework."*

We agree while reading these remarks that the original outline of the manuscript doesn't put enough emphasis on the transient simulation and that it would be better to construct the outline of the paper so as to better echo the title. It is important for us to keep the discussion of the different sensitivity tests. This knowledge is needed to properly analyze the results of the transient simulation and to know what we can or cannot expect from it.  We will add a few results on the transient simulations. But we'll keep most of the content as it is. To better emphasize the results of transient simulations we propose to restructure the manuscript as follow:

1. Introduction
2. Model and experiments
3. Simulated climate and vegetation throughout the mid to late Holocene
4. Multiple vegetation states and uncertainties
5. Conclusion

Compared to the original outline:

1. Introduction
2. Model, mid Holocene and preindustrial experiments
3. Mid-Holocene simulations with interactive vegetation

4. Simulated climate and vegetation throughout the mid to late Holocene
5. Conclusion

The new structure is a response to the reviewer comment to provide a clear framework for the results. The new section 2 will start from the experimental design of the transient experiment; so as to explain that the mid-Holocene is the reference period and only a subset of simulations were run for the pre-industrial period. The discussion of the sensitivity tests will be slightly refocused and redistributed in the different subsections. The discussion on the MH initial vegetation state will be included, but not the discussion on the multi vegetation states for the PI vegetation. The current section 3 on mid-Holocene simulations will thus be redistributed between section 1, and section 4 where a specific focus will be put on the multiple vegetation states for PI and the evaluation of the simulated vegetation for MH and PI using the biomisation method. This is a way to discuss what we call limits in the title. In the new section 3 on the transient simulation we'll slightly enlarge the analysis of the response to the insolation forcing and add a discussion on the climate variables at the regional scales.

The different figures will be reorganized so as to reflect the new outline. It sounds difficult to reduce the number, but we'll find a way to have fewer maps with vegetation changes. It requires some work, but it should be easily done, thanks to the way we organized the model outputs needed to prepare this manuscript.

We also would like to thank reviewer 1 for the list of minor comments that are useful to improve the manuscript.

All the editing comments have been taken into account and already added in the text before any change is made. We provide below some responses for the other comments

*L130: what do you mean with 'transient late Holocene simulation'*

The last 6000 years (I.E from -6000 BP to 0k = 1950 for insolation). This will be stated more clearly in the text

*L243: Please explain the metrics in more detail (e.g. in the Appendix) because the metric package may be unknown to the readers*

A paragraph will be added in the appendix to better explain what is computed

*L314: The heading of chapter 3 is: 'mid-Holocene simulations' so why is there a section dealing with pre-industrial climate?*

We hope it will be less misleading in the new outline. The point is to know how good is the model quite early in the text. In the new version we decided to evaluate the "climate" in section 2 and have the discussion on "vegetation" in section 4. This should better insist on the fact that we have active dynamical vegetation in this simulation and that considering climate or vegetation evaluation can lead to different conclusions on the realism of the simulation depending on the way the evaluation is done.

*L319-320: I do not understand what is meant by 'vegetation biases' in this context. When vegetation is interactive, the calculated vegetation distribution can be biased, but how does this bias impact the representation of the simulated vegetation? Please clarify.*

We only have in mind the biases coming from climate-vegetation feedbacks that amplify the known bias of the model when dynamical vegetation is switch off. We are not in a position where we can tell how the bias in the vegetation model affects the full coupled system.

*L337: What do you mean by this? that the differences in PI simulations are of similar magnitude as the differences between PI and MH ?*

Since the vegetation map are similar in Vmap and Vnone for MH, the difference in PI vegetation between Pi-Vmap and PI-Vnone explains the difference in MH-PI vegetation calculated using the Vmap simulation or the Vnone simulation. We'll revisit the way we discuss it.

*L383: ': : : follow the long term insolation changes in each hemisphere: : :' What about SH Winter? Please be more precise.*

We'll add a discussion on this point, but focusing on the seasonal cycle and the seasonality of the insolation forcing. For the northern regions and the southern hemisphere, part of the answer is in the ocean heat storage and the other part is in the sea-ice and snow cover.

*L409: It seems as if the tree fraction follows the summer insolation change. Please specify and explain. What about the annual mean changes in temperature, precipitation and insolation?*

We will add a short discussion on temperature and precipitation, but for the 3 regions we consider later in the text, considering, min, max and annual mean monthly temperatures and precipitations as well as sea-ice and snow cover for the region north of 60°N and Eurasia.

*L456: What do you mean by 'rapid changes' and if these 'deserve attention' why don't you investigate them in this study?*

We should have included these remarks in the conclusion. It is out of the scope of this paper. So we'll refocus the text.

*L476-477: Is there a possibility to figure out the reasons for having different PI climate-vegetation changes?*

We provide all what we know and the possible caveat in the manuscript. Going further requires a new study and certainly another 1 to 2 years to do it properly with ensemble sensitivity tests. We already checked all what we could check in the last 2 years about it. This is also why it is important for us to show it and discuss it in the manuscript. It can be "by chance" or linked to amplification of small differences in the initial state under modern conditions.

*-L581: Isn't it originally the method of Prentice et al. 2011? What is different to the method of Zhu et al. 2018?*

Yes, the algorithm follows Prentice et al. (2011), with thresholds prescribed as in Zhu et al. (2018). We also tested the different threshold values reported in Figure A2.

We will revise this sentence as: "To convert the modelled PFTs by ORCHIDEE into mega BIOMES, we use the algorithm proposed by Prentice et al. (2011). Figure A2a shows the different threshold values tested in this algorithm, with the black numbers corresponding to the default values used to produce Figure 7 in the main text."

*-L590: It is not obvious why the GDD limit of 500_C is being tested, are these values realistic? I guess ORCHIDEE also uses a GDD limit of > 350_C for the existence of boreal trees vs. tundra (GDD5<350_C). A biomisation using a GDD limit of 500_C thus may not represent the vegetation simulated by the model, because it suggest tundra in regions that are suited for forests.*

The threshold of 500 °C days is tested because it was used in Joos et al. (2004) to convert LPJ-simulated PFT fractions into biome types. ORCHIDEE intrinsically does not use a simple GDD limit to constrain the existence of boreal tree PFTs. GDD thresholds are only used in the phenology module to determine the onset time of leaves, while their values are PFT-specific and are also modulated by the dormancy period, which varies for the same PFT located in different grid cells (see more details in Krinner et al., 2005). By influencing leaf onset, GDD values impact photosynthesis and growth of the PFT, and then indirectly affect establish/mortality rates and finally abundance of this PFT in ORCHIDEE. The biomisaztion algorithm is just a post-processing of the fractional PFT outputs of ORCHIDEE, with some broad-scale empirical thresholds. Therefore, we do not think testing a value of 500 °C days here would be "incompatible" to ORCHIDEE-simulated vegetation.

*-L598-599: Should we now reconsider the choice of bioclimatic limits in the DGVMs?*
*- What about data availability?*

As mentioned above, GDD is not a direct bioclimatic limit inside ORCHIDEE. Furthermore, although changing the GDD limit to 500 °C days improves the metric for tundra in Figure A2b, we should keep in mind that (1) any bias in simulated temperature will also affect the biomisation result and thus the "correctness" compared with the pollen data; and (2) the expansion of tundra over woodland in the case of "GDD=500" compared to "Default" might actually degrade the biome distribution, which cannot be reflected in the "correctness" metric because of limited pollen data in middle Siberia (this is why we mentioned data availability).

*-Fig.4: Why are MH_Vnone and MH_Vmap so different?*

We are not sure we fully understand the question. These simulations start with very different initial state for the land surface model. So it reflects different adjustment time, and the curve show they converge to the same solution. So we would rather say that they are very similar and not different.

*Fig.7: It should be explained, why there is Savanna in the northern latitudes. In my print, the pink and orange color is not really distinguishable. Please state, why there is no grassland in North Africa in your simulation*

As shown in Figure A2a, "savanna and dry woodland" is defined if the foliage projective cover (a combination of simulated fractional coverage and leaf area index) is high but average tree height is not enough. Since tree height is mainly determined by woody biomass in ORCHIDEE, we speculate that a potential underestimation of tree biomass in the model might lead to the replacement of boreal forests with woodlands in the high latitudes. This could be because of bias in climate and/or

bias in ORCHIDEE in terms of photosynthesis or carbon allocation scheme. We will add these discussions in the corresponding text.

For North Africa, the model simulates desert instead of grasslands. This is mainly because of amplification by the climate-vegetation feedback of the underestimated precipitation in this region.

We will change the colors to make them more distinguishable in the revised manuscript.

*-Fig.9: Maybe this figure could be moved to the Appendix*

We will keep this figure in the text and add to it a panel with the seasonal change in incoming solar radiation at TOA in both hemispheres. Showing the forcing we use in the simulation over the 6000 years is important for the discussion. We'll also better emphasize in the text the result of the last period.

*-Fig.10: When looking into palaeo-seasons, one always faces the problem of different calendars. The months NDJF or JJA differ in length between mid-Holocene and PI. It should at least be mentioned in the text and in the caption, that this 'problem' exists and is not considered, neither by the model nor in the analysis. But this problem may change the trends discussed here!*

In practice the effect of calendar over the mid Holocene is small. Joussaume and Braconnot 1997 show it is 5 days at most for the difference in the date of the Autumnal equinox when March 21 is prescribed as the reference date for the vernal equinox in all simulations. We are averaging on long time and show the long term trends, discussing only the significant results. The larger analysis biases resulting from the calendar are found in autumn. We do not discuss this particular season. Our conclusions, given what we are doing here will not be altered by the calendar effect. But we recognize that we need to keep this in mind. For other periods, when eccentricity is larger, this would not be the case.

*-Fig.11: it is 'Northern Hemisphere'. It would also be interesting, how the simulated tree cover and bare-soil fractions at the end of the simulation compare to modern estimates on (natural) tree cover. How large is the underestimation of forest in the high northern latitudes by the model?*

We agree it would be interesting, but we are also concerned that because we do not have land use in the simulations. Land use has an impact at regional scale. However we also know, and this is shown in the MH biome comparisons, that the differences between the simulated vegetation and the real world are larger that differences that would come from land use. Since we decided now to add a biome comparison for PI using pollen data for 0k, we'll consider this remark in the revision. We'll also reinforce the discussion and questions about the evaluation of the vegetation we simulate out of such transient simulation.

*Fig.12: What causes the strong peak around 4.8ka ?*

This event comes from internal noise and/or compound variability events, superimposed on the long term trend induced by the insolation forcing. There is no obvious cause. We checked that it doesn't come from an artificial computing failure when running the simulation.

**Response to Reviewer 2 comments**

Review 2 has very strong comments on the content and the organization of our manuscript.

*"I "recommend modifying the structure and results to better reflect the target audience. You can leave the model development and testing sections, but you need to provide additional context and analyses."*
*« General Comments:*
*The results and analyses left me unsatisfied, especially given the amount of time spent on model testing. Many findings are dismissed as beyond the scope of this paper or for future work. However, without in-depth exploration of at least some of the interesting results of the simulations, the paper feels more like a data description, which is fine, but not especially appropriate for Climate of the Past. I recommend expanding the transient simulation results and analysis, since it is the novel part of this study. There are several topics that could be explored further, such as the importance of dynamic vegetation in the transient climate response (needed for the title), the mechanisms driving multiple equilibria, and comparison with proxy reconstructions. The authors might also want to consider how do these results compare with other transient model simulation?"*

Some of these comments are consistent with those of reviewer 1. We realize that the structure we adopted for this manuscript deserved us. We already provided quite a lot of in depth analyses even though we agree that the section on the transient simulation as it is appears a little bit descriptive. We propose to add a few things on the transient simulation to better discuss the response to the insolation forcing and the linkage between climate and vegetation at the regional scale. But we will keep our initial focus and use the different tests we did to highlight the context in which the simulation can be considered, in particular for future model-data comparisons. This implies that we better highlight the limits we discussed. They come from the possibility of multi-states for vegetation, model biases and caveats for model evaluation on the pre-industrial or the historical period. We thus propose to reorganize the manuscript so as to have the discussion of these points in the last section.  This will allows us to better connect the different pieces and provide a more focused manuscript.

To better emphasize the results of the transient simulations we propose to restructure the manuscript as follow:

6. Introduction
7. Model and experiments
8. Simulated climate and vegetation throughout the mid to late Holocene
9. Multiple vegetation states and uncertainties
10. Conclusion

Compared to the original outline:

6. Introduction
7. Model, mid Holocene and preindustrial experiments
8. Mid-Holocene simulations with interactive vegetation
9. Simulated climate and vegetation throughout the mid to late Holocene
10. Conclusion

As stated above, the new structure is a response to the reviewer request to revisit the structure of the manuscript. The new section 2 will start for the experimental design of the transient experiment; so as to explain that the mid-Holocene is the reference period and that only a subset of simulations was run for the pre-industrial period. The discussion of the sensitivity test will be slightly refocused and redistributed in the different subsections. The construction of the MH initial state for vegetation will also be included, but not the discussion on the possibility for multi vegetation states for PI. The current section 3 on mid-Holocene simulations will thus be redistributed between section 1, and section 4 where a specific focus will be put on the multiple vegetation states for PI and the evaluation of the simulated vegetation for MH and PI using the biomisation method. We'll also emphasize what we call limits in the title. In the new section 3 on the transient simulation we'll slightly enlarge the analysis of the response to the insolation forcing and add a discussion on the climate variables over the three regions.

The different figures will be reorganized so as to reflect the new outline. It sounds difficult to reduce their number, but we'll find a way to have fewer maps with vegetation changes.

Responses to the other comments. The comments dealing with text editing will be considered if still relevant in the revised version of the manuscript. We answer only to questions or to comments considering the content.

*Line 123: Can sea level actually change in the model?*

The ocean model has a free surface. The average sea level evolves with the global surface water budget (evaporation – precipitation – river runoff – water flux from ice sheet). However, the numeric is not designed for sea level large sea level change. It's better to keep it small with regards to the depth of the first level (10 m). The water conservation in the coupled model is thus critical for sea-level stability and to make sure that the sea-level change in a transient experiment is indeed the result of climate changes and not of model spurious drift.

*Line 143: I do not think that this is very good justification for not thoroughly testing modifications against preindustrial climate.*

We do not fully understand this comment. We provide comparison for PI for a subset of simulations. The argument on computing time is only the truth and we had to adjust our strategy to the computing allocation we had. We didn't had enough computing time to run both 1000 years long simulation on MH and PI for all the tests. We started the model developments on PI and then move on MH for the final tests that are presented here that all requested long simulations.

*Lines 152-153: Given the importance of the aerosol responses, why do you prescribe aerosols here? Are dust and sea-salt prescribed to PI? How might this impact climate? I do not find ": : :we also plan to run simulations with fully interactive dust and sea-salt" good justification.*

Here also we do not fully understand this comment. We wrote: because we are developing the interactive version with aerosols that can be run on 6000 year long time periods. What we didn't write is that is also requires developing the full coupling with the interactive vegetation for the dust sources, and that we are here at the first step with dynamical vegetation. It is ongoing work. That would require another 1 to 2 years. Aerosols are set to their pre-industrial values, and they are fully interactive with the radiative code in the atmosphere.

*Line 163: Some of these modifications do not feel robust (e.g. the soil evaporation factor).*
They are, we only show robust results here.

*Lines 175-176: What is the TOA energy imbalance for these runs? This could be important since different simulations are run for different amounts of time. 0.4 Wm-2 is far from zero: : :*

The imbalance is negligible ~0 (with interannual variability around it). Of course, during the adjustment phase, it is equivalent to what is shown at the surface. Part of the small offset at the surface results from small errors when estimating the heat budget from the monthly model outputs. It is not possible from the limited output we kept for the long simulations to properly reallocate the right latent heat values when we are dealing with evaporation or sublimation on surfaces with evolving sea ice or snow. This is done properly during the model run. It is thus mostly a diagnosis error rather than a model imbalance. Note however that interannual variability is of the order of 0.2-0.4 W/m$^2$.

*Line 181: Why does this modification impact the ocean response so dramatically? I thought the hydrologic modification should only impact the land surface. Am I missing something?*

Because we are in a coupled system and that energy is redistributed between land and ocean. Changes in evaporation over land affect moist static energy and its gradients.

*Line 310: how well does 50% compare with other models?*

We will conduct the same biomisation and evaluation against pollen data for PI outputs. This would provide a comparison of model performance in vegetation distribution under different climates. For other models, say PMIP3 outputs, it is difficult because the biomisation algorithm requires some variables (e.g. tree height) that are not usually uploaded in PMIP3. We will, however, add more discussions here about the model-data evaluation, referring to the recent work of Dallmeyer et al., (2018, CPD).

*Lines 325-326: "Since surface variables adjust rapidly, this is a way to compare the rapid adjustment to insolation 326 and the additional effect due to the dynamical vegetation (not discussed here)." Why say this then? It feels like an advertisement: : :*

We agree it is not needed.

*Line 380: Do you mean JJAS? How do you account for the calendar changes?*

We don't account for calendar change. See response to reviewer 1. The changes are limited, even though present over the last 6000 years. Here, we only discuss robust features that would emerge whatever the choice of calendar. We'll add a caution mark on the calendar in the revised version.

*Line 398: Interesting: : :Worth performing spectral analysis on the variability?*

Certainly, we agree, but later and not in this manuscript. It is a subject per se.

*Lines 418-419: Why would this lead to an underestimate?*

Even though the carbon cycle is interactive in the land surface and in the ocean, the fact that the carbon concentration is imposed in the atmosphere in the model prevents carbon feedback between the different reservoirs. Model forced in emission rather than concentrations have a larger range of response in their carbon cycle. This is why we think that in a simulation with where emissions interact with the atmospheric concentration could lead to different results. The wording here was misleading. Underestimated was there to mean not fully computed and that in a fully interactive model the results could be different. We'll revisit the sentence to better reflect what we want to say.

Summary
27/02/2019 17:18:22

Differences exist between documents.

**New Document:**
Braconnotetam_CP-2018-140_vrev
47 pages (3.91 MB)
27/02/2019 17:17:31
Used to display results.

**Old Document:**
cp-2018-140-supplement
44 pages (3.97 MB)
27/02/2019 17:17:30

Get started: first change is on page 1.

No pages were deleted

**How to read this report**

**Highlight** indicates a change.
 indicates deleted content.
🔺 indicates pages were changed.
↔️ indicates pages were moved.

[revised manuscript text omitted]

---

## Referee Report (RR1)

**Comment on „Strength and limits of transient mid to late Holocene simulations with dynamical vegetation" by Pascale Braconnot et al .**

In their manuscript, the authors present the first transient Holocene simulation with the IPSL model, which also contains dynamic vegetation. They discuss the influence of the model setup on the results and work out possible challenges when comparing transient simulations with reconstructions. The authors have once again put a lot of effort into the manuscript and have completely restructured it, which makes the motivation for the extensive discussions much more obvious. The revised version is much clearer and more comprehensible than the first version. The readability has considerably been improved. Nevertheless, I still have a few comments on the general structure:

a) I still find the first part about the different model versions tedious and don't really see the advantage of discussing them so extensively. This does not really fit into this study and the authors do not discussed them further in the course of the article. It is always good to know how changes in the model setup affect the representation of climate and vegetation, and I also think that changes in the model need to be documented and compared with previous versions to better assess the model results, but I think it might as well be shifted to the appendix. Then the focus would be even more on the transient simulation, as the title promises. It would also reduce the number of figures in the main text.

b) The main objective of the study is not fully clear, whether the focus is more on introducing the transient simulation or on the challenges of comparing such simulations with reconstructions, or maybe on both. And the main topic should be reflected in the title. I think the current title is not appropriate, because the strength and limits of transient simulations are only shortly addressed. The 'limits' (or better challenges) are discussed, but most of these limits also exist for time-slice simulations. The „strengths" compared to time-slice experiments should be more strongly emphasised. An important, albeit trivial, aspect in this respect is also that reconstructions usually only exists punctually in large parts of the earth and also in a very coarse temporal resolution. The transient simulations have the great advantage that - as described in the article - they show no fixed climate and vegetation state but also the transition to this state and are therefore (at best) suitable to close the gaps in the reconstructions and may also be used to check the consistency of the records. The records are also not 'perfect' and without biases. With transient simulations more opportunities exist for the model-data comparison.

c) I like the main questions raised in the introduction. The individual chapters are very well aligned to these questions. I think it would increase the clarity of the paper and also 'round off' the paper if these questions were also answered systematically. They could either be picked up (and even repeated) in the conclusion, or  a small summary could be added at the end of each chapter to answer the question and summarize the main, relevant result.

Minor and technical remarks:

- I do not list any spellings or grammar mistakes, these will be fixed during the CP copy-editing.

- L 85: … the timing of the major vegetation change in different regions

- chapter 2.1: If I understand it correctly, the test simulations ran with PMIP3 protocol and the transient simulations with PMIP4 protocol. I still find the large number of simulations confusing, or find it difficult to get what was used in which simulation. Maybe you could extend tables 1 and 2 by

a few columns and thus classify more clearly which model version, which PMIP protocol and which initialization were used. I find the table confusing. For example, actually all simulations with PMIP4 forcing are marked with FPMIP4, but TROHOLV is not, you can only guess the information. For MH-Vmap and MH-Vnone it says that the model ran with 11 layer hydrology and Aerosols and Evaporation factor, for others it doesn't or only indirectly. You could simply add the columns L11, Aer, Ev and then check which components are used, unless this model version has been used in all simulations in table 2, but then you could write it down centrally and not mention it in any simulation. Additionally a column for forcing (PMIP3 or PMIP4) and for initial state (bare soil or map) would be helpful.

L187: Do you mean MH-PMIP3?

L205: it is PI-FPMIP4.

L209-211: Is this sentence complete?

L240: Why have you chosen MH-Vnone as initial state?

L.260: Please define seasonality also in the main text.

L264: I would add (Fig.5) at the end of the sentence.

L271: This statement is too general. According to Fig.6, precipitation and temperature during summer follow the insolation change in the NH, this is also true for NH winter temperature, but not for the other curves.

L277-278: This sentence sounds odd.

L296: only 40% or rather 50%?

L354: there is no substantial reduction of tree cover in Eurasia. These sentences are only valid for the region north of 60°N.

L510-511: I do not understand this sentence.

L530: reconstructions instead of observations

L533: reference of Marsicek et al is missing in the reference list.

L573-574: This sentence sounds odd.

L651-654: The table showing the metrics of the climate-based biomisations only occurs in the Discussion Paper and not in the finally revised version.

Table 2: The heading: Sensitivity experiments to dynamical vegetation …. seems to miss a verb?
Figure 1: ...modified version of the IPSLCM5A-MR model in which …. or something similar.

Figure 3. Description of Panel a) and b) are switched. And ta_200 is probably the temperature on the 200hPa level and not on the 300hPa level.

Figure 4: I wonder if vegetation in MHVnone would converge to  the vegetation in MH-Vnone_FPMIP4. The 'jump' from the grass and tree fraction in MHVnone to MHVnone_FPMIP4 is very large. This is confusing.

Figure 5: The label of the red axis is wrong.

Figure 6, 3rd line: 6000 years BP.

Figure 8: I'm confused why the spread in minimum and maximum temperature increases from 6ka to 0ka for Eurasia and the region north of 60°N.

Figure 10: The last sentence is incomplete.

Figure 11: Please add coordinates for the WAfrican region. And the green curve for north of 60°N is not completely printed on the plot.

Figure 15: The colour bars of the temperature plots are cut off. Line 4: it is c) and d) …. and in line 5 it is TRHOLV.

---

## Author Response (AR2)

Dear Martin,

We revised the manuscript according to the comments of reviewer. We also revisit Fig14 to take into account your remarks on its readability.  We hope that this revision fits the expectations and that you'll find this manuscript in good shape for publication in Climate of the Pascale.

Sincerely

Pascale Braconnot, on behalf of the co-authors.

Response to reviewer 1

First of all we would like to thank the reviewer for the careful reading of the manuscript and the constructive comments. We answer below.

*In their manuscript, the authors present the first transient Holocene simulation with the IPSL model, which also contains dynamic vegetation. They discuss the influence of the model setup on the results and work out possible challenges when comparing transient simulations with reconstructions. The authors have once again put a lot of effort into the manuscript and have completely restructured it, which makes the motivation for the extensive discussions much more obvious. The revised version is much clearer and more comprehensible than the first version. The readability has considerably been improved. Nevertheless, I still have a few comments on the general structure:*

*a) I still find the first part about the different model versions tedious and don't really see the advantage of discussing them so extensively. This does not really fit into this study and the authors do not discussed them further in the course of the article. It is always good to know how changes in the model setup affect the representation of climate and vegetation, and I also think that changes in the model need to be documented and compared with previous versions to better assess the model results, but I think it might as well be shifted to the appendix. Then the focus would be even more on the transient simulation, as the title promises. It would also reduce the number of figures in the main text.*

We fully understand the point made by the reviewer, but we decided to keep section 2.3 in the main text and didn't move it in the appendix. This vision seems also to be shared by reviewer 2 and the editor. We slightly reduced the text were possible. We prefer to only have in the appendix the complement information to the evaluation that would appear redundant in the main text. Even though we agree that this section might appear tedious to reader mainly interested by the transient simulation, we still think it is important to have in the main text the elements to judge how model version and experimental design affect the results.

*b) The main objective of the study is not fully clear, whether the focus is more on introducing the transient simulation or on the challenges of comparing such simulations with reconstructions, or maybe on both. And the main topic should be reflected in the title. I think the current title is not appropriate, because the strength and limits of transient simulations are only shortly addressed. The 'limits' (or better challenges) are discussed, but most of these limits also exist for time-slice simulations. The „strengths" compared to time-slice experiments should be more strongly*

*emphasised. An important, albeit trivial, aspect in this respect is also that reconstructions usually only exists punctually in large parts of the earth and also in a very coarse temporal resolution. The transient simulations have the great advantage that - as described in the article - they show no fixed climate and vegetation state but also the transition to this state and are therefore (at best) suitable to close the gaps in the reconstructions and may also be used to check the consistency of the records. The records are also not 'perfect' and without biases. With transient simulations more opportunities exist for the model-data comparison.*

Yes, the focus is in between the first analyses of the long term trends of transient simulations and the challenges for model data comparison.  We considered the remark on the strength and reinforced some of the ideas in the conclusion. We also modified the title as suggested.

*c) I like the main questions raised in the introduction. The individual chapters are very well aligned to these questions. I think it would increase the clarity of the paper and also 'round off' the paper if these questions were also answered systematically. They could either be picked up (and even repeated) in the conclusion, or a small summary could be added at the end of each chapter to answer the question and summarize the main, relevant result.*

We adopted the first solution and tried to answer more systematically the questions raised in the introduction in the conclusion. Having a summary in each section would lengthen the manuscript that is already long. We therefore reorganize the conclusion to account for points b and c.

*Minor and technical remarks:*
We would like to thank the reviewer for these remarks that are very useful to correct the manuscript.
*- I do not list any spellings or grammar mistakes, these will be fixed during the CP copy-editing.*
*- L 85: … the timing of the major vegetation change in different regions*
corrected

*- chapter 2.1: If I understand it correctly, the test simulations ran with PMIP3 protocol and the transient simulations with PMIP4 protocol. I still find the large number of simulations confusing, or find it difficult to get what was used in which simulation. Maybe you could extend tables 1 and 2 by a few columns and thus classify more clearly which model version, which PMIP protocol and which initialization were used. I find the table confusing. For example, actually all simulations with PMIP4 forcing are marked with FPMIP4, but TROHOLV is not, you can only guess the information. For MH-Vmap and MH-Vnone it says that the model ran with 11 layer hydrology and Aerosols and Evaporation factor, for others it doesn't or only indirectly. You could simply add the columns L11, Aer, Ev and then check which components are used, unless this model version has been used in all simulations in table 2, but then you could write it down centrally and not mention it in any simulation. Additionally a column for forcing (PMIP3 or PMIP4) and for initial state (bare soil or map) would be helpful.*

We added a column as requested to indicate if the forcing is PMIP3 or PMIP4. The model version is already included in Table 1 and we slightly simplified the way it is written in the second column. All simulations with interactive vegetation were run with the final version of the model, so we didn't change this Table, except for the column with the forcing.

*L187: Do you mean MH-PMIP3?*
Yes we do, corrected

*L205: it is PI-FPMIP4.*
Corrected

*L209-211: Is this sentence complete?*
Sentence replaced by : The effect of cloud in the IPSLCM5A-LR simulations results mainly from low level clouds over the ocean (Braconnot and Kageyama, 2015; Vial et al., 2013)

*L240: Why have you chosen MH-Vnone as initial state?*
we added :  because it leads to more realistic forest in the PI-Vnone simulation (see discussion in section 4)

*L.260: Please define seasonality also in the main text.*
Done

*L264: I would add (Fig.5) at the end of the sentence.*
Done

*L271: This statement is too general. According to Fig.6, precipitation and temperature during summer follow the insolation change in the NH, this is also true for NH winter temperature, but not for the other curves.*
We specified that it is for summer in each hemisphere

*L277-278: This sentence sounds odd.*
We replace it by :  SH summer (JJAS) and NH Winter (NDJF) temperatures are both characterized by a first 2000 years warming

*L296: only 40% or rather 50%?*
sorry, yes it is 50%

*L354: there is no substantial reduction of tree cover in Eurasia. These sentences are only valid for the region north of 60°N.*
Eurasia has been suppressed from the sentence

*L510-511: I do not understand this sentence.*
We made two sentences out of it and addes precision. Since there is almost no difference in MH vegetation between Vmap and Vnone, these differences in PI vegetation drive the vegetation differences between MH and PI (Fig. 16).  The MH simulated changes seem larger with Vmap.

*L530: reconstructions instead of observations*
Done

*L533: reference of Marsicek et al is missing in the reference list.*
Corrected

*L573-574: This sentence sounds odd.*
The sentence has been modified : This is in favor of a different equilibrium that is only partially induced by climate-vegetation feedback

*L651-654: The table showing the metrics of the climate-based biomisations only occurs in the Discussion Paper and not in the finally revised version.*
 We changed the reference to:
Dallmeyer, A., Claussen, M. and Brovkin, V.: Harmonizing plant functional type distributions for evaluating Earth System Models, Clim. Past Discuss., 1–51, 2018.
*Table 2: The heading: Sensitivity experiments to dynamical vegetation …. seems to miss a verb?*
*Figure 1: …modified version of the IPSLCM5A-MR model in which …. or something similar.*
Corrected

*Figure 3. Description of Panel a) and b) are switched. And ta_200 is probably the temperature on the 200hPa level and not on the 300hPa level.*
Corrected

*Figure 4: I wonder if vegetation in MHVnone would converge to the vegetation in MH-Vnone_FPMIP4. The 'jump' from the grass and tree fraction in MHVnone to MHVnone_FPMIP4 is very large. This is confusing.*
The jump is not that large between the two. There is a smoothing effect in the curve, so that we miss the last part of Vnone and the first part of _FPMIP4. But it is true that, due to the large oscillation when we start from bare soil, we cannot be 100% sure that we the difference in forcing do not counteract the adjustment of MHVnone.

*Figure 5: The label of the red axis is wrong.*
Corrected

*Figure 6, 3rd line: 6000 years BP.*

*Figure 8: I'm confused why the spread in minimum and maximum temperature increases from 6ka to 0ka for Eurasia and the region north of 60°N.*
For each of the curves we subtracted the 6k values. The curves thus diverge from this origin with time.

*Figure 10: The last sentence is incomplete.*
It is complete in the figure caption. In practice the last words moved to the next page, and we do not know why.

*Figure 11: Please add coordinates for the WAfrican region. And the green curve for north of 60°N is not completely printed on the plot.*
The limits we chose for the drawing explain that the green curve seems to be incomplete. We kept the limits the same in the three plots.

*Figure 15: The colour bars of the temperature plots are cut off. Line 4: it is c) and d) …. and in line 5 it is TRHOLV.*
Thank you for mentioning it. We'll provide the original figure

[revised manuscript text omitted]